# Hydrodynamics with helical symmetry

Jack H. Farrell,[1, *] Xiaoyang Huang,[1, †] and Andrew Lucas[1, ‡]

[1]*Department of Physics and Center for Theory of Quantum Matter,*
*University of Colorado, Boulder CO 80309, USA*
(Dated: August 29, 2022)

We present the hydrodynamics of fluids in three spatial dimensions with helical symmetry, wherein only a linear combination of a rotation and translation is conserved in one of the three directions. The hydrodynamic degrees of freedom consist of scalar densities (e.g. energy or charge) along with two velocity fields transverse to the helical axis when the corresponding momenta are conserved. Nondissipative hydrodynamic coefficients reminiscent of chiral vortical coefficients arise. We write down microscopic Hamiltonian dynamical systems exhibiting helical symmetry, and we demonstrate using kinetic theory that these systems will generically exhibit the new helical phenomena that we predicted within hydrodynamics. We also confirm our findings using modern effective field theory techniques for hydrodynamics. We postulate regimes where pinned cholesteric liquid crystals may possess transport coefficients of a helical fluid, which appear to have been overlooked in previous literature.

## CONTENTS

## 1. INTRODUCTION

The framework of hydrodynamics [1–3] offers a powerful machinery for understanding the late-time and long-distance behavior of many-body classical and/or quantum systems. Indeed, for most ordinary liquids and gases, such as water or air, the famous Navier-Stokes equations arise as the universal effective theory description of long-wavelength dynamics

* jack.farrell@colorado.edu
† xiaoyang.huang@colorado.edu
‡ andrew.j.lucas@colorado.edu

compared to a (nanometer scale) mean free path. These equations arise due to the non-relativistic (approximate) Galilean invariance of the universe.

Over the past few decades in condensed matter physics, hydrodynamics has arisen in a huge number of distinct settings: liquid crystals [4–7], ultracold atomic gases [8–15], phonons in solids [16–20], electron liquids in high-purity solids [3, 21–27], and even active matter such as flocking birds [28–32]. Many of these systems are not Galilean invariant. For example, in liquid crystal suspensions, rotational symmetries are usually spontaneously broken by the alignment of nearby molecules; in the literature, the resulting "order parameter" is often incorporated as a "quasihydrodynamic" degree of freedom with a finite lifetime. However, in electron or phonon liquids in solid-state platforms, it is more appropriate to consider *explicit* breaking of rotational symmetries due to the finite point group of the underlying crystal lattice. Only in recent years has the literature begun to systematically study the consequences of this explicit rotational symmetry breaking [33–39]; many new possible hydrodynamic coefficients and phenomena have already been found.

This work describes the hydrodynamics of three-dimensional systems with the symmetry of a helix. This means that only a linear combination of rotation and translation along a pre-determined axis is a symmetry (see Figure 1). We emphasize that this is conceptually distinct from the spontaneous breaking of translation and rotational symmetries to a helical symmetry in cholesteric liquid crystals [6, 7], since in this system the axis of helical order is spontaneously chosen, *and* there is a propagating Goldstone boson associated to the sliding of the helix. However, considering a system which has the Galilean symmetry group *explicitly* broken down to a helical subgroup is conceptually and technically simpler, even if perhaps it is harder to find in Nature. Moreover, we are not aware of such an approach being taken in prior literature. The purpose of this paper is to emphasize the conceptually important issues that will arise in a helical fluid that do not in other fluids without mixed translation-rotation symmetries.

From a practical perspective, several condensed matter systems could manifest helical order; in particular, we propose later in this work that cholesteric liquid crystals, subject to a certain pinning, will have their symmetry reduced to that of a helix.

Besides the potential for experimental realizations, from a theoretical perspective, understanding hydrodynamics with helical symmetry is worthwhile because it is the simplest symmetry group that preserves a hybrid of a translation symmetry with another "multipolar" symmetry whose density explicitly depends on the spatial coordinates (in this case, the angular momentum density depends on both the spatial coordinate and the momentum density). While the hydrodynamics of multipole-conserving fluids has become a quite active subject recently [40–45], the helical symmetry pattern is interesting both because it is more realizable experimentally, because (as we will show) it removes a hydrodynamic degree of freedom (the momentum density along the helical axis), and because the symmetry contains an explicit length scale $\xi$, which enters the hydrodynamic formalism in some surprising ways. These effects lead to

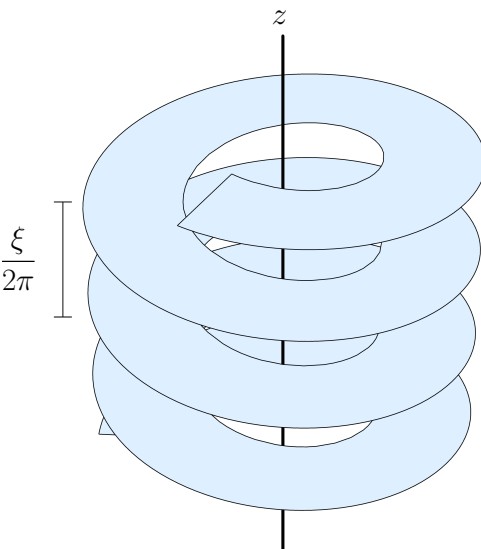

FIG. 1. Cartoon demonstrating helical symmetry. Rotating around the $z$ axis by the angle $\phi$ and correspondingly translating in the $z$ direction by $\xi\phi$ is a symmetry.

some subtlety in the application of existing formalisms, including kinetic theory [2, 46, 47] or effective field theory [48–52], to helical fluids. The purpose of this paper is to solve these issues and to present complementary perspectives on the helical fluid.

In Section 2, we review formally the symmetries relevant to a helical fluid. Using these symmetries, we derive the equations of hydrodynamics based on a conventional Landau perspective in Section 3, and explicitly demonstrate that two (not three) components of momentum represent genuine hydrodynamic modes. Section 4 then demonstrates that new helical transport coefficients will exist in generic microscopic models with helical symmetry, while Section 5 presents a modern effective field theoretic perspective on helical fluids and demonstrates how to write down an effective action for the helical Navier-Stokes equations.

## 2. HELICAL SYMMETRY

The symmetries of a system impose constraints on the hydrodynamic equations. This work studies a three-dimensional helical fluid that conserves the following charges: the number of particles $N$, the two components of momentum in a particular plane (taken to be $P_x, P_y$), the "twist momentum"

$$K_z \equiv P_z + \xi^{-1} L_z, \tag{2.1}$$

where $L_z = xP_y - yP_x$ is the $z$ component of angular momentum, and the total energy $E$. In particular, translations in the $x$–$y$ plane, generated by $P_x$ and $P_y$, leave the fluid invariant, but translations in the $z$ direction and rotations about the $z$ axis do not—except in the combination $K_z$. All other continuous symmetries, including rotations and boosts, are explicitly broken.

Of course, one could consider additional scalar charges, and this would not effect our conclusions. For example, we neglect the effects of energy conservation on the equations of motion in this work until Sec. 5. Since energy is another scalar charge, any new symmetry-allowed contributions coming from the helix will be captured by our analysis of $N$—including $E$ offers nothing new, from this perspective. Consequently, we will omit $E$ in the following analysis.

The "line group" (subgroup of all symmetries that can leave a single line fixed) is isomorphic to O(2) – the representations of this group will constrain the invariant tensors we can include in our hydrodynamic equations. To show this isomorphism, first consider continuous rotations in the $x$–$y$ plane parameterized by

$$\begin{pmatrix} x \\ y \\ z \end{pmatrix} \to \begin{pmatrix} \cos\theta & -\sin\theta & 0 \\ \sin\theta & \cos\theta & 0 \\ 0 & 0 & 1 \end{pmatrix} \begin{pmatrix} x \\ y \\ z \end{pmatrix}. \tag{2.2}$$

(These are accompanied by translations which we do not consider here.) The top left $2 \times 2$ block here is naturally denoted as $R_{AB}$ and is the two-dimensional vector representation of SO(2). Here and below, lower case latin indices $i, j \ldots = (x, y, z)$ stand for spatial coordinates, while upper case latin indices $A, B \ldots = (1, 2)$ only label the two dimensional subspace, namely the $x$–$y$ plane. One can easily see that all symmetry generators are covariant under this transformation and map to other (linear combinations of) symmetry generators Next, consider a mirror reflection, $\sigma$, in the $y$–$z$ plane, taken without loss of generality to be:

$$\begin{pmatrix} x \\ y \\ z \end{pmatrix} \to \begin{pmatrix} 1 & 0 & 0 \\ 0 & -1 & 0 \\ 0 & 0 & -1 \end{pmatrix} \begin{pmatrix} x \\ y \\ z \end{pmatrix}. \tag{2.3}$$

Note that $\sigma \cdot K_z = -K_z$, which is still a symmetry generator. We do not need to change the sign of $\xi$ under this transformation. Lastly, all of our symmetry generators have definite transformations under time-reversal $\Theta$:

$$\Theta \cdot (E, N, P_A, K_z) = (E, N, -P_A, -K_z). \tag{2.4}$$

Hence, we will further impose microscopic time-reversal symmetry. $R$, $\sigma$ and $\Theta$, together with $P_{x,y}$, make up the spacetime symmetries of the helical fluid.

Let us focus on the spatial symmetries alone, consisting of the rotations $R_{AB}$ and the mirror reflection $\sigma$. The three dimensional coordinate vector space $(x, y, z)$ forms a three-dimensional representation of the symmetry group (we have explicitly presented it above). The group itself is isomorphic to O(2). We can see this by noting that the $(x, y)$ components of the vector space form a two-dimensional representation of the symmetry group, as $R_{AB}$ and $\sigma$ are block diagonal. With a slight abuse of notation, we then observe the group composition law

$$R(\theta)\sigma = \sigma R(-\theta). \tag{2.5}$$

This means that the group is non-Abelian; moreover, it is the semidirect product $\text{SO}(2) \rtimes \mathbb{Z}_2 = \text{O}(2)$.[1]

There are exactly three inequivalent building-block tensors which we can construct that are $\text{O}(2)$-invariant, and out of which all other invariant tensors can be built via tensor products: they are $\delta_{ij}$, the completely antisymmetric tensor $\epsilon_{ijk}$, and finally the 'upper left' block of the identity tensor, $\delta_{AB}$ (implicitly, we take its $zz$-component to be 0). That these are the only invariant tensors can be seen as follows. First, imagine we have full rotation symmetry so that the spatial coordinates furnish a representation of $\text{SO}(3)$. Then there are two invariant tensors, $\epsilon_{ijk}$ and $\delta_{ij}$. If we fully broke $\text{SO}(3)$ to $\text{SO}(2)$ (only rotations around $z$ are allowed), then we would find an additional invariant tensor: the unit vector $e_z$ pointing in the $z$ direction. Note that this would make $\delta_{AB}$ and $\epsilon_{AB} = \epsilon_{ABz}$ into invariant tensors as well. However, as mentioned earlier, the helical symmetry contains one extra symmetry, the reflection $\sigma$, under which $e_z \to -e_z$ is not invariant. Only the tensor $e_z \otimes e_z$ is invariant. We can equivalently state this as the fact that $\delta_{AB}$ is a new invariant tensor for the helical symmetry group. These invariant tensors will play an extensive role in the remainder of the paper.

## 3. HYDRODYNAMICS

Having established the symmetries of our theory, we now write down the general form of the hydrodynamic equations that describe the distribution of conserved quantities $P_x$, $P_y$, and $N$ over space. Hydrodynamics is an effective theory that models the evolution of densities (corresponding to conserved quantities) at late times and long wavelengths compared to the mean free path; accordingly, if $\ell$ is a microscopic length scale for the system, $e.g.$ a scattering length, the quantity $(\ell \boldsymbol{\nabla})$ is small. We may then expand the current and stress tensor in powers of $\ell \boldsymbol{\nabla}$, and in this paper we will keep first order terms. Note that one can always adopt the "Landau fluid frame" in which only spatial gradient corrections are included, and we will adopt that convention in this paper.

### 3.1. Gradient expansion

Landau's procedure for deriving hydrodynamic equations involves writing down the most general equations of motion consistent with the problem's symmetries. While this procedure can apparently lead to erroneous results (e.g. results that are not compatible with a fluctuation-dissipation theorem) in certain exotic fluids [50], it will turn out to be correct for the helical fluid (as we will confirm in Section 5). Consider first the scalar $N$ written as the integral of a space and time dependent charge density $n$:

$$N = \int \mathrm{d}\mathbf{x}\, n(\mathbf{x}, t). \tag{3.1}$$

For $N$ to be a conserved quantity, we require that $\mathrm{d}N/\mathrm{d}t = 0$, which by the divergence theorem requires that $\partial_t n$ differs from 0 only by the divergence of some vector $\mathbf{J}(\mathbf{x}, t)$ that is a function of the hydrodynamic degrees of freedom. This results in a *continuity equation*:

$$\partial_t n + \partial_i J_i = 0. \tag{3.2}$$

For the helical fluid, we will produce continuity equations for the scalar charge $n$ as well as the in-plane components of momentum density, $\pi_A$. The $z$-momentum $P_z$ is *not* a conserved quantity, so we will not give an equation for $\pi_z$; the helical fluid only flows in the $x$ and $y$ directions. Section 5 explains the disappearance of $\pi_z$ in detail. In terms of these variables, we find continuity equations:

$$\partial_t n + \partial_i J_i = 0, \tag{3.3a}$$
$$\partial_t \pi_A + \partial_i \tau_{iA} = 0. \tag{3.3b}$$

Our program, then, is to determine the most general form of the stress tensor $\tau_{iA}$ and current $J_i$ as functions of $n$ and $\pi_i$. Finally, we note that at linear order it is conventional to replace $\boldsymbol{\pi}$ with its thermodynamic conjugate, the velocity $\mathbf{v}$, related to momentum by $\boldsymbol{\pi} = n_0 \mathbf{v}$. To be clear, we are working at linear order in $\mathbf{v}$, but we have no need to specify

---

[1] The equals sign here denotes group isomorphism. The isomorphism is essentially by construction, as one notes that the general group element $M$ as a $3 \times 3$ matrix consists of an arbitrary $2 \times 2$ matrix $S_{AB} \in \text{O}(2)$ in the top left corner ($M_{AB} = S_{AB}$), with $M_{zz} = \det(M)$ and $M_{zA} = M_{Az} = 0$.

to the linear regime for $n$—this explains why we keep the variable $n$ in our equations rather than a perturbation $\delta n$ from the steady state. That way, starting with $\tau_{iA}$ we can write down its general form as:

$$\tau_{iA} = A_{iA}n + B_{iAB}v_B + C_{iAk}\partial_k n + D_{iAkB}\partial_k v_B + \ldots \tag{3.4}$$

The coefficients $A, B, C, D$ must be built from the O(2)-invariant tensors given in Section 2. Since we are in a nonlinear regime for $n$, the coefficients could depend on $n$ in general.

First we study the allowed terms with zero derivatives, corresponding to the $A$ and $B$ tensors above. The only contribution to $A_{iA}$ is:

$$A_{iA} = mc^2 \delta_{iA}, \tag{3.5}$$

where $c$ is the speed of sound in the fluid. This term corresponds to the ordinary hydrostatic pressure in linear response, $P(n) = c^2 n$.

Next, consider the $B$ terms. In general, since ideal hydrodynamics cannot manifestly violate time reversal symmetry, no terms linear in $v_x, v_y$ may appear. Thus the coefficients in $B$ all vanish identically.

Now we turn to the one-derivative corrections parameterized by $C$ and $D$ in Eq. (3.4), and in particular, we begin with the coefficients $D$. Conventionally, the term $D$ is called the viscosity tensor. In the absence of external forces, we expect terms in the equation of motion proportional to gradients of velocity to correspond to frictional effects. Then the standard analysis gives one contribution to $D$, the isotropic viscosity tensor:

$$D_{iAkB} = \eta \left( \delta_{ik}\delta_{AB} + \delta_{iB}\delta_{Ak} - \delta_{iA}\delta_{kB} \right) + \zeta \delta_{iA}\delta_{kB}. \tag{3.6}$$

Here, $\eta$ and $\zeta$ are called bulk and shear viscosity respectively. However, in the helical fluid, the viscosity tensor need not be fully isotropic; the special $z$ direction means it must only be invariant under rotations SO(2) rotations rather than $SO(3)$. The result is that one additional viscosity component is allowed in the helical fluid:

$$D_{iAkB} = -\eta_z \delta_{iz}\delta_{kz}\delta_{AB}. \tag{3.7}$$

Because the term with this coefficient is proportional to $\partial_z$, this contribution to the stress tensor indeed vanishes in the case of uniform rotation around the $z$ axis, as a frictional effect should.

So far we have only used two of the invariant tensors, $\delta_{ij}$ and $\delta_{AB}$. Using $\epsilon_{ijk}$ gives only one additional piece:

$$D_{iAkB} = \eta_\circ \epsilon_{iA}\epsilon_{kB} \tag{3.8}$$

The coefficient $\eta_\circ$ is called rotational viscosity. It turns out, however, that the helical fluid with only $x, y$ velocities cannot have rotational viscosity. The reason is that the vanishing of $v_z$ (since it is not a hydrodynamic degree of freedom) implies angular momentum is approximately conserved:

$$\begin{aligned} 0 \approx \frac{\mathrm{d}L_z}{\mathrm{d}t} &= \int \mathrm{d}^3 x \left( x\pi_y - y\pi_x \right) \\ &= -\int \mathrm{d}^3 \left( x\partial_i \tau_{iy} - y\partial_i \tau_{ix} \right) \\ &= \int \mathrm{d}^3 x \left( \tau_{xy} - \tau_{yx} \right). \end{aligned} \tag{3.9}$$

Then $\tau_{xy} - \tau_{yx}$ should be a total derivative:

$$\tau_{AB}\epsilon_{ABz} = \partial_k V_{kz} \tag{3.10}$$

for some function $V_{kz}$. Helical symmetry requires $V_{kz} = \xi^{\pm 1} F \delta_{kz}$ for a scalar function $F$, so that $\tau_{xy} - \tau_{yx}$ must be proportional to a $z$ gradient:

$$\tau_{xy} - \tau_{yx} = \xi^{\pm 1} \partial_z F \tag{3.11}$$

We have pulled out a factor of $\xi^{\pm 1}$ to give $\tau_{AB}$ the correct parity transformation properties. This result prohibits rotational viscosity in the helical fluid.

Finally, we turn to the coefficient $C_{ijk}$, and it is here that we find novel effects arising from helical symmetry—there exist tensors that are *not* invariant under the group of 3-dimensional rotations, $SO(3)$, but *are* invariant under the helix group $O(2)$. One option is:

$$C_{ijk} = mc^2 \left( \alpha \epsilon_{ijz}\delta_{kz} + \beta \epsilon_{izk}\delta_{jz} \right). \tag{3.12}$$

where $\alpha$ and $\beta$ are, at this level, phenomenological coefficients that are calculable at least in principle from the kinetic theory of gases, and we have pulled out a factor of $mc^2$ for later convenience. Note that these terms are compatible with the requirement that the antisymmetric part of the stress tensor be a total $z$ derivative. Strictly, another symmetry-allowed contribution is $\epsilon_{kij}\delta_{kz}$ However, in the equations of motion, this term features the non-hydrodynamic mode $v_z$, which vanishes. As we will see, these two terms are nondissipative. These two effects correspond to additional terms in the stress tensor:

$$\tau_{AB} = \alpha\epsilon_{AB}\partial_z\mu, \tag{3.13a}$$

$$\tau_{zA} = \beta\epsilon_{AB}\partial_B\mu, \tag{3.13b}$$

where $\mu = n/(\rho c^2)$ is the thermodynamic conjugate of $n$ and $\rho = mn_0$ is the momentum susceptibility, $\pi = \rho v$, and $n_0$ is the equilibrium density. Notice that these terms transform correctly under the reflection from Sec. 1—that is, the equations of motion for $\pi_i$ are all covariant under the symmetry group of the helix. In practice, it is this condition that is easier to enforce when trying to produce the correct equations of motion.

Having described the form of the stress tensor, we now turn to the particle number current $J_i$. The current may as well be expanded in derivatives, and again we find two additional terms allowable by symmetry. The additional contributions from helical symmetry are:

$$J_z = a\epsilon_{AB}\partial_A v_B - \sigma'\partial_z\mu, \tag{3.14a}$$

$$J_A = b\epsilon_{AB}\partial_z v_B - \sigma\partial_A\mu, \tag{3.14b}$$

where $a$ and $b$ are again phenomenological coefficients. We have also included an incoherent diffusivity $\sigma, \sigma'$ which can have different coefficients in the $xy$-plane and in the $z$-direction.

The contributions to $\tau_{ij}$ and $J_i$, however, are not independent—instead, they are in general related by thermodynamic constraints. The theorem, due to Onsager [33, 50], is best expressed by relating the currents $\tau_{ij}$ and $J_i$ to the gradients of the dynamical variables via the matrix $\kappa$. Next, we assemble all the 'currents' into a matrix $\mathcal{J}_\phi^i$, corresponding to the $i^{\text{th}}$ spatial component of the current for the variable $\phi = (n, \pi_x, \pi_y, \pi_z)$, and we define also abstractly the thermodynnamic conjugates of each charge, $\mu_\phi$. Explicitly, this gives:

$$\mathcal{J}_\phi^i = \kappa_{\phi\psi}^{ij} \, \partial_j\mu_\psi, \tag{3.15}$$

where there is a sum over $j$ and also over $\psi = (n, \pi_x, \pi_y, \pi_z)$. $\kappa$ can be thought of as a matrix with two indices as long as $(i, \phi)$ and $(h, \psi)$ are grouped. The statement is that the block-diagonal matrix $\kappa$ is either symmetric or antisymmetric within each diagonal block—that is:

$$\kappa_{\phi\psi}^{ij} = \pm\kappa_{\psi\phi}^{ji}. \tag{3.16}$$

The correct signature $\pm$ can be determined as follows. First, consider the effect of time-reversal $\Theta$ on the helical fluid. $\Theta$ is a symmetry of the helix with $\mathbf{P} \to -\mathbf{P}$, $K_z \to -K_z$, and $N \to +N$. To preserve Eq. (3.16), we must require $\kappa$ be unchanged under the combination of $\Theta$ with interchanging $\phi$ and $\psi$.

$$(\Theta)_\phi\kappa_{\psi\phi}^{ij}(\Theta)_\psi = \kappa_{\phi\psi}^{ij}, \tag{3.17}$$

where there is no sum over $\phi$ and $\psi$. Now, we specialize to $\phi = \mu$ and $\psi = \mathbf{v}$, recalling $\mathbf{v}$ is odd under $\Theta$ while $\mu$ is even. Then:

$$\alpha\epsilon_{AB} = \kappa_{v_B\mu}^{Az} = -\kappa_{\mu v_B}^{zA} = -a\epsilon_{AB} \implies \alpha = -a, \tag{3.18a}$$

$$\beta\epsilon_{AB} = \kappa_{v_z\mu}^{BA} = b\epsilon_{AB} \implies \beta = -b. \tag{3.18b}$$

These two additional *nondissipative* contributions to $\tau_{ij}$ and their 'partners' in $J_i$ encode the effects of the helical symmetry on the hydrodynamic equations, the consequences of which we will explore in the following sections.

### 3.2. Relaxation of $z$-velocity

In this section, we provide an explanation for the relaxation of $v_z$, which we have assumed in the previous section; indeed, we calculate its relaxation rate. Here, we write a quasihydrodynamic equation for $\pi_z$ along with hydrodynamic ones for the true conserved quantities:

$$\partial_t n + \partial_i J_i = 0, \tag{3.19a}$$

$$\partial_t \pi_A + \partial_i \tau_{iA} = 0, \tag{3.19b}$$

$$\partial_t \pi_z + \partial_i \tau_{iz} = -\Gamma(n, \boldsymbol{\pi}). \tag{3.19c}$$

Here, the source term $\Gamma(n, \boldsymbol{\pi})$ included in the equation of motion for $\pi_z$ parameterizes the fact that $\pi_z$ is not conserved.

Since $\pi_z$ is nonvanishing, conservation of $K_z$ implies angular momentum $L_z$ is not conserved; as a result, rotational viscosity is allowed in this theory. By symmetry considerations, then, we should include the following term in the stress tensor:

$$\tau_{xy} - \tau_{yx} = -2\eta_\circ \left( \partial_x v_y - v_y \partial_x \right) \equiv -2\eta_\circ \omega_z. \tag{3.20}$$

However, given the helical symmetry, the above "guess" for the rotational viscosity term must be modfied as follows. Recall that, according to the hydrodynamic ansatz [3], the helical fluid is at equilibrium with the density matrix:

$$\rho_{\text{eq}} = \frac{\exp(-\beta \left( H - \mu N - v_A P_A - w K_z \right))}{\mathcal{Z}} \tag{3.21}$$

where $w$ is the thermodynamic conjugate of $K_z$, having units of velocity. In a reference frame where $w$ is sourced (though note this is *not* a Galilean boost!), we might expect

$$v_z \rightarrow v_z + w \tag{3.22a}$$

$$\Omega_z \rightarrow \Omega_z + w/\xi. \tag{3.22b}$$

The shift in $v_z$ arises because $v_z$ is the thermodynamic conjugate variable to $P_z$, so it should shift simply under $w$. Analogously, angular velocity $\Omega_z$ is thermodynamically conjugate to angular momentum $L_z$. Of course, $\Omega_z$ is not a genuine hydrodynamic mode, but it does modify

$$v_x \rightarrow v_x - \Omega_z y, \tag{3.23a}$$

$$v_y \rightarrow v_y + \Omega_z x. \tag{3.23b}$$

Under such a change of reference frame, the vorticity $\omega_z = \partial_x v_y - \partial_y v_x$ transforms nontrivially:

$$\omega_z \rightarrow \omega_z + 2\xi^{-1} w. \tag{3.24}$$

But the vorticity $\omega_z$ appears in a dissipative term in the equations of motion, namely the term parameterized by rotational viscosity $\eta_\circ$—the generalized "boost" along the helix will then create extra entropy. We conclude that, rather than simply $\omega_z$, the rotational viscosity must couple instead to the "boost" invariant combination $\omega_z - 2\xi^{-1} v_z$.. Accordingly, the stress tensor's rotational viscosity term is:

$$\tau_{xy} - \tau_{yx} = -2\eta_\circ \left( \omega_z - 2\xi^{-1} v_z \right). \tag{3.25}$$

The inclusion of a term proportional to $v_z$ in $\tau_{xy} - \tau_{yx}$ causes $v_z$ to be a relaxing degree of freedom. To start, we may compute the rate of change of total momentum in the $z$ direction as:

$$\begin{aligned}
\frac{\mathrm{d}P_z}{\mathrm{d}t} &= -\xi^{-1} \frac{\mathrm{d}L_z}{\mathrm{d}t} \\
&= -\xi^{-1} \int \partial_t (x\pi_y - y\pi_x) \\
&= -\xi^{-1} \int \left( \tau_{xy} - \tau_{yx} \right). \tag{3.26}
\end{aligned}$$

Thus any term in $\tau_{xy} - \tau_{yx}$ that is *not* a total derivative will contribute to the non-conservation of $P_z$. Looking back at our construction of $\tau_{ij}$, the only term in $\tau_{xy} - \tau_{yx}$ that is not a total derivative is the novel piece from rotational viscosity:

$$\tau_{xy} - \tau_{yx} \sim 4\frac{\eta_\circ}{\xi} v_z. \tag{3.27}$$

Then, removing the integral sign in Eq. (3.26), we find the equation of motion for $\pi_z$ is:

$$\partial_t \pi_z + (\text{total derivatives}) = -\xi^{-1}(\tau_{xy} - \tau_{yx}) = -\frac{4\eta_\circ}{\rho_\parallel \xi^2} v_z + \frac{2\eta_\circ}{\xi} \omega_z, \tag{3.28}$$

where $\rho_\parallel$ is the momentum susceptibility in the $z$ direction. The $z$ velocity decays with relaxation time

$$\gamma^{-1} = \frac{\rho_\parallel \xi^2}{4\eta_\circ}. \tag{3.29}$$

As a result, on long time scales $t \gg \gamma^{-1}$, we may treat $v_z \sim 0$. When we treat the helical fluid in the language of kinetic theory, we will come to a different perspective on the relaxation with the same conclusion. That the rotational viscosity term features the combination $\omega_z - 2\xi^{-1}v_z$ rather than simply $\omega_z$ actually guarantees that, after $v_z$ has relaxed, the rotational viscosity decouples from the equations of motion for $v_x$ and $v_y$. This can be guessed since $\omega_z$ always appears in combination with the nonhydrodynamic mode $v_z$. To see the disappearance of $\eta_\circ$ in more detail, though, let us return to (3.28) and notice that the term in brackets on the left hand side is simply the usual $n_0 \partial_z \mu$ up to $O(\boldsymbol{\nabla}^2)$. Since the right hand side is $-\xi^{-1}(\tau_{xy} - \tau_{yx})$, after a long time (when $\partial_t \pi_z \approx 0$), we find:

$$\tau_{xy} - \tau_{yx} = -n_0 \xi \partial_z \mu, \tag{3.30}$$

giving

$$\alpha = -n_0 \frac{\xi}{2}. \tag{3.31}$$

The antisymmetric part of the stress tensor is *fixed* to be proportional to $\partial_z \mu$. This remarkable fact, that helical symmetry eliminates rotational viscosity and completely prescribes the value of $\alpha$, will be justified more rigorously with kinetic theory (Section 4) and effective field theory (Section 5).

### 3.3. Equations of motion

The result of the previous discussions, after $v_z$ has relaxed, is a set of hydrodynamic equations for $n, v_x, v_y$. Omitting bulk viscosity $\zeta$ for conciseness, and omitting $\eta_\circ$ for reasons described above, they are:

$$n_0 \partial_t v_A + (\alpha + \beta)\, \epsilon_{BA} \partial_B \partial_z \mu = -n_0 \partial_A \mu + m^{-1}(\eta \partial_i \partial_i v_A + \eta_z \partial_z^2 v_A), \tag{3.32a}$$

$$\partial_t n + n_0 \partial_i v_i - (\alpha + \beta)\, \epsilon_{AB} \partial_A \partial_z v_B = D \partial_A \partial_A n + D' \partial_z^2 n, \tag{3.32b}$$

where we have recalled the sign difference in coefficients between the helical part of the current and the stress tensor, and we have also defined $\rho$ as the momentum susceptibility in the plane ($\pi = \rho v$). We have also defined the incoherent diffusion constants $D = \sigma/\chi$ and $D' = \sigma'/\chi$, where $\chi$ is the charge susceptibility. There is also a quasihydrodynamic equation for $v_z$, which we will write omitting $\eta_\circ$ and $\zeta$:

$$n_0 \partial_t v_z = -c^2 \partial_z n + \eta m^{-1} \partial_i \partial_i v_z - n_0 \gamma v_z. \tag{3.33}$$

As a final comment, we note that the terms corresponding to $\alpha$ and $\beta$ are reminiscent of the hydrodynamic terms that appear in situations where the 'chiral vortical effect' is relevant [53].

One way to understand Eqs. (3.32) is to study the quasinormal modes, perturbations to the steady uniform state proportional to $\exp(i(\mathbf{k} \cdot \mathbf{x} - \omega t))$. In that case (3.32) becomes a matrix equation:

$$\begin{pmatrix} -D(k_x^2 + k_y^2) - D'k_z^2 - i\omega & (-\tilde{\alpha}k_y k_z + in_0 k_x) & (\tilde{\alpha}k_x k_y + in_0 k_y) & in_0 k_z \\ c^2 n_0^{-1} ik_x + c^2 n_0^{-1}\tilde{\alpha}k_y k_z & \tilde{\eta}m^{-1}n_0^{-1}k^2 - i\omega & 0 & 0 \\ c^2 n_0^{-1} ik_y - c^2 n_0^{-1}m^{-1}\tilde{\alpha}k_x k_z & 0 & \tilde{\eta}m^{-1}n_0^{-1}k^2 - i\omega & 0 \\ c^2 n_0^{-1} ik_z & 0 & 0 & \eta n_0^{-1}m^{-1}k^2 - i\omega + \gamma \end{pmatrix} \begin{pmatrix} \delta n \\ \delta v_x \\ \delta v_y \\ \delta v_z \end{pmatrix} = \begin{pmatrix} 0 \\ 0 \\ 0 \\ 0 \end{pmatrix} \tag{3.34}$$

where we have defined the combination $(\alpha + \beta) \equiv \tilde{\alpha}$ and also $\tilde{\eta} = \eta + \eta_z$. Let us additionally define angle dependent effective diffusion constants and viscosities:

$$\tilde{D}(\theta) = D \sin^2 \theta + D' \cos^2 \theta, \tag{3.35a}$$

$$\tilde{\eta}(\theta) = \eta \sin^2 \theta + \eta_z \cos^2 \theta. \tag{3.35b}$$

With these definitions, solving the eigenvalue equation for $\omega$ gives $\omega(\mathbf{k})$ for the four eigenmodes up to the third order in powers of $k$:

$$\omega(\mathbf{k}) \approx \begin{cases} -\mathrm{i}n_0^{-1}m^{-1}\tilde{\eta}k^2, \\ -c\sin\theta k - \mathrm{i}k^2\left(\dfrac{c^2\cos^2(\theta)}{2\gamma} + \dfrac{1}{2}\tilde{D}(\theta) + \dfrac{\tilde{\eta}(\theta)}{2mn_0}\right) - f(\theta)k^3, \\ +c\sin\theta k - \mathrm{i}k^2\left(\dfrac{c^2\cos^2(\theta)}{2\gamma} + \dfrac{1}{2}\tilde{D}(\theta) + \dfrac{\tilde{\eta}(\theta)}{2mn_0}\right) + f(\theta)k^3, \\ -\mathrm{i}\gamma + \dfrac{\mathrm{i}k^2\left(c^2mn_0\cos^2\theta + \gamma\eta\right)}{\gamma}. \end{cases} \tag{3.36}$$

Here, we have defined the coefficient of the $k^3$ term as $f(\theta)$, which is the first term that involves the novel coefficient $\tilde{\alpha}$ and has the explicit form. Of course, to take the $k^3$ terms seriously would require deriving the equations of motion to third order in gradients, which, while straightforward, is unwieldly. The discussion above suffices to show the lowest order contribution of $\alpha$ on the dispersion of the normal modes. The explicit form of $f(\theta)$ is:

$$f(\theta) = f^{(0)}(\theta) + \tilde{\alpha}^2\frac{c}{n_0}(\sin\theta - \sin^3\theta), \tag{3.37}$$

where we have defined:

$$f^{(0)}(\theta) = \frac{1}{\sin\theta}\left(-\frac{c^3}{8\gamma^2} - \frac{\tilde{D}(\theta)^2}{8c} + \tilde{D}(\theta)\left(\frac{c}{4\gamma} - \frac{\tilde{\eta}(\theta)}{4cmn_0}\right) - \frac{\tilde{\eta}(\theta)^2}{8cm^2n_0^2} + \frac{c\tilde{\eta}(\theta)}{4\gamma mn_0}\right)$$

$$+ \left(\frac{3c^3}{4\gamma^2} - \frac{c\tilde{D}(\theta)}{4\gamma} - \frac{c\tilde{\eta}(\theta)}{4\gamma mn_0}\right)\sin(\theta) - \frac{5c^3}{8\gamma^2}\sin^3\theta. \tag{3.38}$$

Thus $f^{(0)}$ is the form of this term when $\tilde{\alpha} = 0$. We note that the $\tilde{\alpha}$ correction does not represent any *new* angular dependence in the quasinormal mode spectrum — it just renormalizes the coefficients of $\sin\theta$ and $\sin^3\theta$ already present in $f^{(0)}$. The subtle effect in terms of the quasinormal modes will make detecting the $\tilde{\alpha}$ term difficult by simply studying the dynamics of a helical fluid.

### 3.4. Generation of torque via an electric field

An alternative and "useful" consequence of the additional term in $\tau_{AB}$ is the generation of torque via an external electric field.

Imagine an object with cross-sectional area $A$ in the helical fluid, and consider applying a uniform electric field $\mathbf{E} = E\hat{\mathbf{z}}$. This is equivalent to establishing a gradient in the chemical potential:

$$\partial_z\mu = -eE. \tag{3.39}$$

The torque applied by the fluid on the object can be computed from the asymmetric part of the stress tensor. Remember that the components of force (per unit length) exerted by the fluid on a line element oriented with unit vector $\hat{\mathbf{n}}$ are $F_A = \tau_{AB}n_B dl$, and the torque is correspondingly $\Gamma_z = \epsilon_{AB}x_AF_B$. Because of the contraction with $\epsilon_{AB}$, only the novel asymmetric part of $\tau_{AB}$ will contribute. We can therefore calculate the torque as:

$$\begin{aligned} \Gamma_z &= \oint_C \epsilon_{AB}x_A\tau_{BC}n_C \, dl, \\ &= \oint_C \epsilon_{AB}x_A(\alpha\epsilon_{BC}\partial_z\mu)n_C \, dl, \\ &= \alpha eE \oint_C x_An_A \, dl, \\ &= 2\alpha eEA, \end{aligned} \tag{3.40}$$

where in the last line we used the divergence theorem. The generation of torque proportional to $E$ in linear response offers one signature of the helical fluid.

More generally, we will see in (5.39) that a *pressure gradient* can generically cause this torque. Hence even in an uncharged fluid, where the conserved U(1) is mass and not electric charge, this effect may be detectable in experiment.

## 4. MICROSCOPIC HAMILTONIAN DYNAMICS AND KINETIC THEORY

The construction of hydrodynamics as an effective theory from the Landau perspective suffices to formulate the equations correctly. Still, for deeper intuition about the origin of the additional effects and to verify that there is no subtler physics that forces the coefficients to vanish, it helps to derive the hydrodynamic equations from kinetic theory for weakly-interacting particles in a system with helical symmetry. In this section, we will first write down a general single-particle Hamiltonian whose kinetic energy indeed has only helical symmetry. Then, we will consider an ensemble of weakly-interacting particles with this helical kinetic energy, applying the methods of kinetic theory to establish that the coefficients $\alpha$ and $\beta$ are both non-zero.

### 4.1. Helical kinetic terms

Before analyzing a gas of particles, we need present a single particle Hamiltonian that has helical symmetry. To this end, we look for modifications to the usual kinetic energy $\mathbf{p}^2/2m$ such that $K_z$ is a conserved quantity:

$$\{H, K_z\} = 0.$$

Defining $p_x = p_\perp \cos\theta, p_y = p_\perp \sin\theta$, such a Hamiltonian must be of the following form:

$$H = H(p_\perp, p_z, \theta - z/\xi). \tag{4.1}$$

It is helpful to work with a simple quadratic Hamiltonian:

$$H = \frac{p^2}{2m} + \frac{1}{2m^\star}\left(p_x \cos\left(\frac{z}{\xi}\right) + p_y \sin\left(\frac{z}{\xi}\right)\right)^2, \tag{4.2}$$

which can also be written as:

$$H = \frac{p_\perp^2 + p_z^2}{2m} + \frac{p_\perp^2}{2m^\star}\cos^2\left(\theta - \frac{z}{\xi}\right) \tag{4.3}$$

Here, $m^\star$ is a parameter with units of mass, and we have defined $p_x = p_\perp \cos\theta, p_y = p_\perp \sin\theta$. The velocity is $\mathbf{v} = \dot{\mathbf{x}} = \partial H/\partial \mathbf{x}$. Note that the limit $m^\star \to \infty$ corresponds to the restoration of full rotational invariance.

The single-particle trajectories offer some insight into the origin of the term parameterized by $\alpha$ in the hydrodynamic equations of motion—the idea is that motion in the $z$ direction naturally leads to the production of angular momentum. To understand this simply, notice that conservation of $K_z$ requires

$$\frac{dL_z}{dt} = -\xi \frac{dp_z}{dt}, \tag{4.4}$$

which means any production of $p_z$ must also create (negative) angular momentum in the $z$ direction. In the context of a single particle moving in the helical Hamiltonian, $p_x$ and $p_y$ are conserved quantities, constants, that are set by the initial conditions. For given values of $p_\perp$ and $\theta$, then, the helix contribution is a periodic potential, and the problem may be understood as a one-dimensional problem for $z(t)$. The trajectories relevant for transport consist of unbound motion in the $z$ direction and spiralling motion in the $x$ and $y$ directions: to be clear, the velocity in the azimuthal $\hat{\boldsymbol{\theta}}$ direction is $v_\theta = -p_\perp^{-1}\,\partial H/\partial\theta$. But $dL_z/dt = \partial H/\partial\theta$, so we have:

$$v_\theta = \frac{\xi}{p_\perp}\frac{dp_z}{dt}. \tag{4.5}$$

Acceleration in the $z$ direction necessarily accompanies spiral motion. We will see in a later section that it is precisely this fact, that Hamiltonian evolution naturally converts $p_z$ into $L_z$, that gives rise to the term we call $\alpha$ in our hydrodynamic equations.

### 4.2. Kinetic theory

We now turn to the collective properties of a gas of weakly interacting particles with the Hamiltonian $H$, which we will analyze using methods of kinetic theory of gases and in particular the linear algebra formalism of [47]. To begin,

recall that the Boltzmann equation for the distribution function $f$ admits equilibrium thermal solutions of the form $f_0(\mathbf{x}, \mathbf{p}) = f_0(H)$ where $f_0$, is any function of the Hamiltonian. To take deviations from equilibrium into account, we may expand the distribution function in a Taylor series as:

$$f(\varepsilon) = f_0(\varepsilon) - \frac{\partial f_0}{\partial \varepsilon} \Phi(\mathbf{x}, \mathbf{p}). \tag{4.6}$$

Following [47], we now define a vector $|\Phi\rangle$ representing the perturbation to the distribution function $\Phi$:

$$|\Phi\rangle \equiv \int d\mathbf{p} \, dz \, \Phi(\mathbf{x}, \mathbf{p}) \, |\mathbf{p}, z\rangle, \tag{4.7}$$

where the basis vectors $|\mathbf{p}, z\rangle$ have an inner product:

$$\langle \mathbf{p}, z | \mathbf{p}', z' \rangle = \frac{\delta(\mathbf{p} - \mathbf{p}')\delta(z - z')}{(2\pi\hbar)^3 \xi} \left( -\frac{\partial f_0}{\partial \varepsilon} \right). \tag{4.8}$$

We may then write the Boltzmann equation as:

$$\partial_t |\Phi\rangle + \mathsf{L}_0 |\Phi\rangle + \mathsf{W} |\Phi\rangle = 0, \tag{4.9}$$

where $\mathsf{W}$ is the linearized collision integral and $\mathsf{L}_0$ is the streaming operator:

$$\mathsf{L}_0 = \frac{\partial H}{\partial p_i} \frac{\partial}{\partial x_i} - \frac{\partial H}{\partial z} \frac{\partial}{\partial p_z}. \tag{4.10}$$

$\mathsf{W}$ encodes the decay of vectors that do not correspond to conserved quantities as a result of scattering from other particles, and $\mathsf{L}_0$ gives the ordinary Hamiltonian or "streaming" evolution.

We now apply this formalism to compute the transport coefficients, focusing first on the novel ones $\alpha$ and $\beta$. To that end, we first define vectors corresponding to the various objects and currents in our theory. For the calculation of $\alpha$, the relevant objects are the antisymmetric part of the in-plane stress tensor $\tau_{xy} - \tau_{yx}$ and the velocity in the $z$ direction $v_z$. Letting $\Theta \equiv \tau_{xy} - \tau_{yx}$ for brevity, we define the corresponding vectors in the linear algebra language as:

$$|\Theta\rangle = \int d\mathbf{p} \, dz \, (v_x p_y - v_y p_x) \, |\mathbf{p}, z\rangle, \tag{4.11}$$

$$|J_z\rangle = \int d\mathbf{p} \, dz \, v_z \, |\mathbf{p}, z\rangle, \tag{4.12}$$

with $\mathbf{v} = \partial H / \partial \mathbf{p}$ computed from the microscopic Hamiltonian.

We now explain how the $\alpha$ and $\beta$ terms arise as first order derivative corrections to hydrodynamics—for details on the full construction of zeroth and first order hydrodynamics from kinetic theory, see *e.g.* [33]. To calculate hydrodynamics at first order in derivatives, we write the Boltzmann equation formally as a vector equation with two subspaces. The first block corresponds to the *slow* degrees of freedom; in other words, it is the subspace spanned by vectors $|\chi\rangle$ such that $\mathsf{W} |\chi\rangle = 0$, namely vectors corresponding to conserved quantities $n$, $\pi_A$. We will denote by $|\Phi\rangle_s$ the components of $|\Phi\rangle$ in this subspace. We then find:

$$\partial_t \begin{pmatrix} |\Phi\rangle_s \\ |\Phi\rangle_f \end{pmatrix} + \begin{pmatrix} \mathsf{L}_0^{ss} & \mathsf{L}_0^{sf} \\ \mathsf{L}_0^{fs} & \mathsf{L}_0^{ff} \end{pmatrix} \begin{pmatrix} |\Phi\rangle_s \\ |\Phi\rangle_f \end{pmatrix} + \begin{pmatrix} 0 & 0 \\ 0 & \mathsf{W}_{ff} \end{pmatrix} \begin{pmatrix} |\Phi\rangle_s \\ |\Phi\rangle_f \end{pmatrix} = 0 \tag{4.13}$$

The second subspace labelled by $f$ corresponds to the *fast* modes that relax after a short time $\gamma^{-1}$. Assuming $\partial_t |\Phi\rangle_f \approx 0$ on large time scales, we find:

$$|\Phi\rangle_f = - \left( \mathsf{L}_0^{ff} + \mathsf{W}_{ff} \right)^{-1} \mathsf{L}_0^{fs} |\Phi\rangle_s \tag{4.14}$$

That way, we find an effective Boltzmann equation for $|\Phi\rangle_s$:

$$\partial_t |\Phi\rangle_s + \mathsf{L}_0^{ss} |\Phi\rangle_s - \mathsf{L}_0^{sf} \left( \mathsf{L}_0^{ff} + \mathsf{W}_{ff} \right)^{-1} \mathsf{L}_0^{fs} |\Phi\rangle_s = 0. \tag{4.15}$$

The last term contains the derivative corrections to hydrodynamics. To see this, recall that the streaming operator $\mathsf{L}_0$ from Eq. (4.10) has two terms. The first contains gradients $\partial_i$, while the second contains the momentum gradient in

the $z$ direction $\partial/\partial p_z$. We are now operating in the $|\Phi\rangle_s$ subspace, and our hydrodynamic ansatz is that $\Phi$ depends only on *fast* modes: not $\pi_z$. That way, only the terms in $\mathsf{L_0}^{fs}$ and $\mathsf{L_0}^{sf}$ with gradients survive. Explicitly, then, dropping the labels $f$, $s$, our Boltzmann equation for the slow modes becomes:

$$\partial_t |\Phi\rangle_s + \mathsf{L_0} |\Phi\rangle_s + \mathbf{v} \cdot \frac{\partial}{\partial \mathbf{x}} (\mathsf{L_0} + \mathsf{W})^{-1} \mathbf{v} \cdot \frac{\partial}{\partial \mathbf{x}} |\Phi\rangle_s = 0. \tag{4.16}$$

Taking the inner product of both sides of this equation with $\epsilon_{AB} \langle \pi_B|$ and integrating by parts, we find $\alpha$ may be computed as a matrix element of the operator $(\mathsf{W} + \mathsf{L_0})^{-1}$ between $|\Theta\rangle$ and $|J_z\rangle$:

$$-2\alpha = \langle \Theta | (\mathsf{W} + \mathsf{L_0})^{-1} | J_z \rangle. \tag{4.17}$$

The above matrix element depends on both the streaming operator $\mathsf{L_0}$ and the collision matrix $\mathsf{W}$. Accordingly, we should establish the action of these operators on the vectors of interest.

Let us begin by considering how $\mathsf{W}$ acts. Recall the form of the Boltzmann equation:

$$\partial_t |\Phi\rangle + \mathsf{L_0} |\Phi\rangle + \mathsf{W} |\Phi\rangle = 0. \tag{4.18}$$

Let us define the vector associated with the density of $K_z$, namely $|k_z\rangle = |p_z\rangle + \xi^{-1} |\ell_z\rangle$, where $|\ell_z\rangle = x |p_x\rangle - y |p_x\rangle$. We may write the rate of change of the total $K_z$ as:

$$\frac{\mathrm{d}K_z}{\mathrm{d}t} = \int \mathrm{d}\mathbf{x}\, \partial_t \langle k_z | \Phi \rangle = -\int \mathrm{d}\mathbf{x}\, \langle k_z | \mathsf{L_0} | \Phi \rangle - \int \mathrm{d}\mathbf{x}\, \langle k_z | \mathsf{W} | \Phi \rangle. \tag{4.19}$$

Integrating by parts, we find that under the $\mathbf{x}$ integral sign, $\mathsf{L_0}$ is antisymmetric, and we therefore find:

$$\int \mathrm{d}\mathbf{x}\, \langle k_z | \mathsf{L_0} | \Phi \rangle = -\int \mathrm{d}\mathbf{x}\, \langle \Phi | \mathsf{L_0} | k_z \rangle,$$
$$= \frac{1}{\xi} \int \mathrm{d}\mathbf{x}\, \langle \Phi | \Theta \rangle,$$
$$= 0. \tag{4.20}$$

The last equality follows because the single-particle Hamiltonian conserves $K_z$, meaning $\mathsf{L_0} |k_z\rangle = 0$. As a result, we conclude that in order for $\mathrm{d}K_z/\mathrm{d}t = 0$, we must have:

$$\mathsf{W} |k_z\rangle = 0. \tag{4.21}$$

On the other hand, we clearly have $\mathsf{W} |\ell_z\rangle = 0$ because $\mathsf{W} |p_A\rangle = 0$. We therefore obtain:

$$\mathsf{W} |p_z\rangle = 0. \tag{4.22}$$

In other words, $\mathsf{W}$ annihilates both $|\ell_z\rangle$ and $|p_z\rangle$, even though neither of these quantities are conserved. It turns out that the *streaming* dynamics is responsible for the non-conservation of these quantities rather than the collision integral.

Next, we may also directly calculate the action of the helical streaming operator on the vector $|J_z\rangle$, noticing that $\partial_z H = -\xi^{-1}\partial_\theta H$:

$$\mathsf{L_0} |J_z\rangle = \int \mathrm{d}\mathbf{p}\, \mathrm{d}z\, |\mathbf{p}, z\rangle \left( v_z \frac{\partial}{\partial z} - \frac{\partial H}{\partial z} \frac{\partial}{\partial p_z} \right) \frac{p_z}{m}$$
$$= \int \mathrm{d}\mathbf{p}\, \mathrm{d}z\, |\mathbf{p}, z\rangle \frac{1}{\xi} \frac{\partial H}{\partial \theta} \frac{1}{m}$$
$$= -\frac{1}{m\xi} |\Theta\rangle. \tag{4.23}$$

Armed with these results, we are ready to compute the coefficient $\alpha$. Our strategy will be to calculate the matrix $(\mathsf{W} + \mathsf{L_0})^{-1}$ approximately by first projecting $(\mathsf{W} + \mathsf{L_0})$ onto the subspace spanned by $|p_z\rangle$ and $|\Theta\rangle$. We begin by defining normalized versions of the vectors:

$$|\tilde{\Theta}\rangle = \frac{|\Theta\rangle}{\sqrt{\langle \Theta | \Theta \rangle}},$$

$$|\tilde{p_z}\rangle = \frac{|p_z\rangle}{\sqrt{\langle p_z|p_z\rangle}}.$$

Assuming $\mathsf{W}|\Theta\rangle = \gamma|\Theta\rangle$ (relaxation time approximation), we have:

$$(\mathsf{L_0} + \mathsf{W})|\tilde{p_z}\rangle = \sqrt{\frac{\langle\Theta|\Theta\rangle}{\langle p_z|p_z\rangle}}\frac{|\tilde{\Theta}\rangle}{\xi},$$

$$(\mathsf{L_0} + \mathsf{W})|\tilde{\Theta}\rangle = -\sqrt{\frac{\langle\Theta|\Theta\rangle}{\langle p_z|p_z\rangle}}\frac{|\tilde{p_z}\rangle}{\xi} + \gamma|\tilde{\Theta}\rangle. \tag{4.24}$$

Since the vectors $|\tilde{p_z}\rangle$ and $|\tilde{\Theta}\rangle$ are orthonormal, we may write $(\mathsf{L_0} + \mathsf{W})$ as:

$$(\mathsf{L_0} + \mathsf{W}) = \sqrt{\frac{\langle\Theta|\Theta\rangle}{\langle p_z|p_z\rangle}}\begin{pmatrix} 0 & \dfrac{1}{\xi} \\ -\dfrac{1}{\xi} & \sqrt{\dfrac{\langle p_z|p_z\rangle}{\langle\Theta|\Theta\rangle}}\gamma \end{pmatrix}. \tag{4.25}$$

Inverting, this $2\times2$ matrix, we find, in this basis:

$$(\mathsf{L_0} + \mathsf{W})^{-1} \approx \xi^2\sqrt{\frac{\langle p_z|p_z\rangle}{\langle\Theta|\Theta\rangle}}\begin{pmatrix} \sqrt{\dfrac{\langle p_z|p_z\rangle}{\langle\Theta|\Theta\rangle}}\gamma & -\dfrac{1}{\xi} \\ \dfrac{1}{\xi} & 0 \end{pmatrix}. \tag{4.26}$$

Remember that we are interested in computing $\langle\Theta|(\mathsf{L_0} + \mathsf{W})^{-1}|J_z\rangle$. Since $|J_z\rangle = |p_z\rangle/m$, we can immediately read this matrix element off as:

$$-2\alpha \approx \frac{\xi}{m}\langle p_z|p_z\rangle. \tag{4.27}$$

The coefficient $\alpha$, then, is clearly nonvanishing, and is also nondissipative. In fact, in this approximation, the rate of relaxation of $\tau_{xy} - \tau_{yx}$, which is $\gamma$, does not come in to the calculation of $\alpha$—the latter is a derivative coefficient coming purely from the nontrivial streaming operator $\mathsf{L_0}$. In contrast, the coefficients of other derivative corrections to hydrodynamics arise from matrix elements of $\mathsf{W}$. Moreover, we recognize $|p_z\rangle = m|J_z\rangle$, and we also know that $\langle J_z|p_z\rangle = n_0$, the equilibrium density, always. Accordingly we can write:

$$\alpha = -\frac{n_0\xi}{2}. \tag{4.28}$$

The value of $\alpha$ agrees with what we predicted in Section 3. Notice that this result is agnostic to the microscopic details—it requires only that $H$ depends nontrivially on the combination $\theta - z/\xi$.

Having established the origin of $\alpha$ and determined its approximate value, we should next calculate the coefficient $\beta$. Like $\alpha$, $\beta$ is given by a certain matrix element of $(\mathsf{L_0} + \mathsf{W})^{-1}$:

$$\beta = -\langle\tau_{yz}|(\mathsf{L_0} + \mathsf{W})^{-1}|J_x\rangle. \tag{4.29}$$

The kets $|\tau_{yz}\rangle$ and $|J_x\rangle$ are defined as:

$$|\tau_{yz}\rangle = \int \mathrm{d}^3\mathbf{p}\,\mathrm{d}z\,|\mathbf{p}, z\rangle\left(\frac{p_x}{m} + \frac{p_\perp}{m^\star}\sin(z/\xi)\cos(\theta - z/\xi)\right)p_z \equiv |\tau_{yz}^{(0)}\rangle + |\tau_{yz}^{(1)}\rangle, \tag{4.30}$$

$$|J_x\rangle = \int \mathrm{d}^3\mathbf{p}\,\mathrm{d}z\,|\mathbf{p}, z\rangle\left(\frac{p_y}{m} + \frac{p_\perp}{m^\star}\sin(z/\xi)\cos(\theta - z/\xi)\right) \equiv |J_x^{(0)}\rangle + |J_x^{(1)}\rangle\Big). \tag{4.31}$$

and in the last line we have defined $|\tau_{yz}^{(0)}\rangle$ and $|\tau_{yz}^{(1)}\rangle$ to be the "ordinary" part (coming from $\mathbf{p}^2/2m$ part of $H$) of the stress tensor and the unusual part (from the $m^\star$ term), with analagous definitions for $|J_x\rangle$ and $|J_y\rangle$. It may be checked that these two contributions are orthogonal, that is $\langle\tau_{yz}^{(0)}|\tau_{yz}^{(1)}\rangle = 0$.

In the case of $\beta$, the strategy employed to calculate $\alpha$ is not necessary since the collision integral does not annihilate either $|\tau_{yz}\rangle$ or $|J_x\rangle$. As a result, in the relaxation time approximation $\mathsf{W}|J_x\rangle \sim \gamma|J_x\rangle$, we may employ a simpler approximation. Expanding $(\mathsf{W} + \mathsf{L}_0)^{-1}$ in powers of $\gamma^{-1}$ gives:

$$(\mathsf{W} + \mathsf{L}_0)^{-1} \approx \mathsf{W}^{-1} - \mathsf{W}^{-1}\mathsf{L}_0\mathsf{W}^{-1} + \dots , \tag{4.32}$$

which leads to a first order approximation for $\beta$:

$$\beta \approx \langle \tau_{yz}|\mathsf{W}^{-1}|J_x\rangle \approx \gamma^{-1}\langle \tau_{yz}|J_x\rangle = 0. \tag{4.33}$$

This contribution vanishes due to the parity of the $p_z$ integral. Then we should move on to the second order correction:

$$\beta = -\gamma^{-2}\langle \tau_{yz}|\mathsf{L}_0|J_x\rangle . \tag{4.34}$$

Evaluating this term requires working out the action of $\mathsf{L}_0$ on $|J_x\rangle$. We have:

$$
\begin{aligned}
\left| v_z \frac{\partial}{\partial z}(v_x) \right\rangle &= \left| \frac{p_z}{m}\frac{\partial}{\partial z}\frac{1}{m^\star}\cos(z/\xi)\,(p_x\cos(z/\xi) + p_y\sin(z/\xi)) \right\rangle \\
&= \left| \frac{p_z}{m}\frac{1}{m^\star}\left(-\xi^{-1}\sin(z/\xi)\,(p_x\cos(z/\xi) + p_y\sin(z/\xi)) - \xi^{-1}\cos(z/\xi)\,(p_x\sin(z/\xi) + p_y\cos(z/\xi))\right) \right\rangle \\
&\equiv -\frac{1}{\xi}|\tau_{yz}^{(1)}\rangle + |a\rangle
\end{aligned}
\tag{4.35}
$$

The action of $\mathsf{L}_0$ on $|J_x\rangle$ gives exactly the ket $|\tau_{yz}^{(1)}\rangle$, up to an additional piece we have called $|a\rangle$. However, it can also be shown that $\langle a|\tau_{xy}^{(0)}\rangle = 0$. That way, we arrive at an expression for $\beta$ in kinetic theory:

$$\beta \approx \frac{\langle \tau_{yz}^{(1)}|\tau_{yz}^{(1)}\rangle}{\gamma^2} \sim \frac{1}{(\gamma m^\star)^2}. \tag{4.36}$$

Thus $\beta$ is also nonvanishing in Kinetic theory.

Next, we may compute the various effective viscosities and compare their magnitudes. Ordinarily, the entries in the viscosity tensor are given by matrix elements of $\mathsf{W}^{-1}$, with the assumption that $\mathsf{W} \ll \mathsf{L}_0$. However, since the helical theory has a contribution to $\mathsf{L}_0$ that is not surpressed by a derivative, the "effective" viscosity coefficients arise from the full operator $(\mathsf{L}_0 + \mathsf{W})^{-1}$. The four nonvanishing viscosity coefficients are then:

$$\eta = \frac{\langle \tau_{xy} + \tau_{yx}|(\mathsf{L}_0 + \mathsf{W})^{-1}|\tau_{xy} + \tau_{yx}\rangle}{4}, \tag{4.37a}$$

$$\zeta = \frac{\langle \tau_{xx} + \tau_{yy}|(\mathsf{L}_0 + \mathsf{W})^{-1}|\tau_{xx} + \tau_{yy}\rangle}{4}, \tag{4.37b}$$

$$\eta_z = \frac{\langle \tau_{zx}|(\mathsf{L}_0 + \mathsf{W})^{-1}|\tau_{zx}\rangle}{4} = \eta + O(1/(m^\star)^2) \tag{4.37c}$$

$$\eta_\circ^{\text{eff}} = \frac{\langle \Theta|(\mathsf{L}_0 + \mathsf{W})^{-1}|\Theta\rangle}{4} = 0. \tag{4.37d}$$

In the case of rotational viscosity $\eta_\circ^{\text{eff}}$, we were able to compute the value exactly using (4.26). Note that this calculation verifies that rotational viscosity does not appear in the helical theory, as we had predicted on general grounds in Section 3.

Finally, we may compute the incoherent diffusivities $D$ and $D'$. To this end, let us first define the incoherent currents:

$$\left| J_i^{\text{inc}} \right\rangle \equiv |J_i\rangle - \frac{\langle \pi_i|J_i\rangle}{\langle \pi_i|\pi_i\rangle}|\pi_i\rangle . \tag{4.38}$$

Note that in the microscopic model we have chosen, $\left| J_z^{\text{inc}} \right\rangle = 0$, but it could be nonvanishing in a more complicated model where $H$ has a term like $p_z^4$ (still consistent with helical symmetry). For our purposes, it is sufficient to use the ordinary current $|J_z\rangle$. The incoherent diffusivities are then:

$$D \sim \left\langle J_x^{\text{inc}}|\mathsf{W}^{-1}|J_x^{\text{inc}}\right\rangle \frac{c^2}{m}, \tag{4.39}$$

$$D' \sim \langle J_z | (\mathsf{W} + \mathsf{L_0})^{-1} | J_z \rangle . \tag{4.40}$$

Notice that (4.26) fixes the value of $D'$ as (in the limit of large $\xi$ where we expect (4.26) to hold):

$$D' \approx \frac{\xi^2 n_0^2}{\eta_\circ} . \tag{4.41}$$

Finally, the derivation of $\alpha$ and $\beta$ in kinetic theory also offers a different perspective on the Onsager reciprocity arguments described in an earlier section. We have shown that calculating $\alpha$ and $\beta$, which are coefficients of the corresponding terms in $\tau_{ij}$, requires determining the matrix elements $\langle \Theta | \mathsf{L_0} | J_z \rangle$ and $\langle \tau_{yz} | \mathsf{L_0} | J_x \rangle$ respectively. In the same way, we could directly calculate the coefficients of the 'partner' terms appearing in $J_i$; these coefficients simply correspond to matrix elements of the adjoint operator, e.g. $\langle J_z | \mathsf{L_0} | \Theta \rangle$. Recall the form of $\mathsf{L_0}$:

$$\mathsf{L_0} = \frac{\partial H}{\partial p_i} \frac{\partial}{\partial x_i} - \frac{\partial H}{\partial z} \frac{\partial}{\partial p_z} . \tag{4.42}$$

In this theory, $\partial_z H$ does not depend on $p_z$. Additionally, $\mathsf{L_0} F(H) = 0$ holds for any function $F$ of the Hamiltonian $H$, so integrating by parts gives:

$$\langle \psi | \mathsf{L_0} | \phi \rangle = - \langle \phi | \mathsf{L_0} | \psi \rangle \tag{4.43}$$

for any functions $\psi$, $\phi$. Thus $\mathsf{L_0}$ is antihermitian, and the coefficients in $J_i$ should differ from their partners in $\tau_{ij}$ by a sign.

## 4.3. Emergent 2D hydrodynamics again

The kinetic theory also offers a different and more transparent perspective on the relaxation of $v_z$ and its relation to the rotational viscosity $\eta_\circ$—indeed, it also clarifies the origin of the value of $\alpha$. To start, by taking an inner product of the Boltzmann equation with $\langle p_z |$, we may calculate the evolution of $\pi_z$ as:

$$\partial_t \langle p_z | \Phi \rangle = - \langle p_z | \mathsf{L_0} | \Phi \rangle ,$$
$$= -n_0 \partial_z \mu - \frac{1}{\xi} \langle \Theta | \Phi \rangle + O(\boldsymbol{\nabla}^2) . \tag{4.44}$$

This indeed show that $\pi_z$ obeys a hydrodynamic equation up to a source term that is precisely $\xi^{-1}(\tau_{xy} - \tau_{yx})$ Next, let us consider the antisymmetric part of the stress tensor, $\tau_{xy} - \tau_{yx} \equiv \Theta = \langle \Theta | \Phi \rangle$. Since this quantity has zero overlap with the hydrodynamic modes, we may assume the collision integral relaxes $| \Theta \rangle$:

$$\mathsf{W} | \Theta \rangle \approx \gamma | \Theta \rangle . \tag{4.45}$$

Then, using the Boltzmann equation, we may calculate:

$$\partial_t \Theta + \langle \Theta | \mathsf{L_0} | \Phi \rangle = -\gamma \langle \Theta | \Phi \rangle . \tag{4.46}$$

After a long time scale compared to $\gamma^{-1}$, we expect $\langle \Theta | \Phi \rangle$ to relax to a steady state. At that point, we will have $\tau_{yx} - \tau_{yx} = -\gamma \langle \Theta | \mathsf{L_0} | \Phi \rangle$. Computing the latter gives:

$$\langle \Theta | \mathsf{L_0} | \Phi \rangle = \frac{1}{2} \langle \Theta | \Theta \rangle \omega_z - \langle \Theta | \frac{\partial H}{\partial z} \frac{\partial}{\partial p_z} | p_z \rangle v_z$$
$$= \left( \frac{1}{2} \omega_z - \frac{v_z}{\xi} \right) \langle \Theta | \Theta \rangle \tag{4.47}$$

But recall that the rotational viscosity $\eta_\circ$ (not $\eta_\circ^{\text{eff}}$!) is given by:

$$4\eta_\circ = \langle \Theta | \mathsf{W}^{-1} | \Theta \rangle \approx \gamma^{-1} \langle \Theta | \Theta \rangle . \tag{4.48}$$

The last line is again the relaxation time approximation. The above result lets us write $\langle \Theta | \Theta \rangle$ in terms of $\eta_\circ$ and $\gamma$, giving:

$$\langle \Theta | \mathsf{L_0} | \Phi \rangle \approx \left( -\frac{1}{2} \omega_z + \frac{v_z}{\xi} \right) \gamma \langle \Theta | \mathsf{W}^{-1} | \Theta \rangle$$

$$= 4\gamma\eta_\circ \left( \frac{v_z}{\xi} - \frac{\omega_z}{2} \right). \tag{4.49}$$

This way, we conclude that:

$$\tau_{xy} - \tau_{yx} = \langle \Theta | \Phi \rangle = -4\eta_\circ \left( \frac{v_z}{\xi} - \frac{\omega_z}{2} \right) \tag{4.50}$$

Finally, then, we arrive at an equation of motion for $\pi_z \equiv \langle p_z | \Phi \rangle$:

$$\partial_t \pi_z + n_0 \partial_z \mu + O(\boldsymbol{\nabla}^2) = -\frac{4\eta_\circ}{\xi^2 \rho_\parallel} \pi_z + \frac{2\eta_\circ}{\xi} \omega_z \tag{4.51}$$

This shows that $\pi_z$ relaxes at rate $\Gamma = 4\eta_\circ/\xi^2 \rho_\parallel$. Note that the relaxation rate computed here agrees with that derived in Section 3. Because we now see that $\pi_z$ is relaxing, we may postulate a steady state where $\partial_t \pi_z = 0$. Then we find the antisymmetric part of the stress tensor is fixed:

$$\frac{(\tau_{xy} - \tau_{yx})}{\xi} = -\frac{4\eta_\circ}{\xi^2 \rho_\parallel} \pi_z + \frac{2\eta_\circ}{\xi} \omega_z = -n_0 \partial_z \mu \tag{4.52}$$

Accordingly, $\tau_{yx} - \tau_{yx} = -n_0 \xi \partial_z \mu$, giving $\alpha = -n_0 \xi/2$, recovering the result of both our guess from Section 3 and our explicit calculation from kinetic theory.

Finally, with the lessons from Section 3 and Section 4, we may interrogate the range of validity of helical hydrodynamics and discuss its subtle limits. At first glance, in the toy model we have discussed, it appears that there are two possible paths toward restoring full Galilean symmetry: we may send $\xi \to \infty$, or we may send $m^\star \to \infty$. Both limits however, seem problematic; for one, the coefficient $\alpha$ does not depend on $m^\star$. On the other hand, $\alpha$ is proportional to $\xi$, which is especially troubling given that $\alpha$ has a clear physical effect, namely the torque on a body immersed in the helical fluid—a large $\xi$ would imply an unphysically large torque. The solution to both of these concerns is that helical hydrodynamics breaks down. To see why, recall that helical symmetry gives the following time scale for the relaxation of $\pi_z$:

$$\tau = \frac{\rho \xi^2}{4\eta_\circ}, \tag{4.53}$$

in agreement with (3.29). Additionally, we notice from (4.48) that $\eta_\circ \sim 1/(m^\star)^2$ in this theory. Accordingly, the time scale on which $\pi_z$ relaxes is roughly:

$$\tau \sim (m^\star \xi)^2, \tag{4.54}$$

since the other quantities do not depend on $m^\star$ or $\xi$. But helical hydrodynamics, and in particular the $\alpha$ term, only appear *after* $\pi_z$ has relaxed to a steady state. It is therefore clear that in the limits where $m^\star \to \infty$ or $\xi \to \infty$ hydrodynamics is not a valid theory—we would have to wait increasingly long times to see helical hydrodynamics emerge. In other words, in either of the limits that restore Galilean symmetry, helical hydrodynamics, inherently a low-frequency theory, simply does not apply.

## 5. EFFECTIVE FIELD THEORY

In this section, we develop a hydrodynamic effective field theory for the helical fluid following [48–50]. This approach will allow us to confirm the validity of the Landau approach for this type of fluid.

### 5.1. Overview of the approach

Consider a generating function in the Schwinger-Keldysh (S-K) formalism [54] for correlators of a conserved U(1) current $J^\mu$ and the energy and momentum tensor $T_\alpha^\mu$ ($s = 1, 2$)

$$e^{W[e_{s,\mu}^\alpha, A_{s,\mu}]} = \text{tr} \left( \rho_0 U^\dagger(e_{2\mu}^\alpha, A_{2\mu}) U(e_{1\mu}^\alpha, A_{1\mu}) \right), \tag{5.1}$$

where $\rho_0$ is the thermal density matrix, $U$ denotes the unitary time evolution operators, and $e^\alpha_\mu$, $A_\mu$ are the background vielbein and gauge field, respectively. In the flat spacetime limit and to the first order in $e^\alpha_\mu$, we can explicitly think of $U$ as

$$U(e^\alpha_\mu, A_\mu) = \exp\left[i \int dt d^d x \ \left(e^\alpha_\mu(x) T^\mu_\alpha(x) + A_\mu(x) J^\mu(x)\right)\right]. \tag{5.2}$$

Here and below, Greek $\mu, \nu, \cdots = 0, x, y, z$ indices will represent physical spacetime indices, while $i, j, \cdots$ represent only spatial components. $\alpha, \beta, \ldots$ represent spacetime vielbein indices, while $b, c, \ldots$ represent spatial vielbein indices only (we reserve $a$ to describe $a$-fields in the Keldysh contour!). Vielbein indices are raised and lowered with a flat Minkowski metric. We see that the vielbein $e^\alpha_\mu$ are effectively sources for the stress tensor $T^\mu_\alpha$. Unlike most of the literature, but following [50], we strongly emphasize that with low spatial symmetry, it is helpful to think of $T^\mu_\alpha$ as carrying two different types of indices: the $\mu$ index reminds us that this is a current, while $\alpha$ encodes the energy/momentum component which is being conserved. There need not in general be any symmetry under exchanging $\mu$ and $\alpha$. We will frequently use *primed* indices $b', c', \ldots$ to denote the $x/y$ vielbein coordinates alone. Computing the full generating function from the microscopic dynamics is generically difficult, so we introduce a path integral formalism with an effective action that would arise from integrating out all of the non-hydrodynamic modes. We can simply include, in the spirit of effective field theory, any term which is consistent with our postulated symmetries.

Following [48], we define Stueckelberg fields $X^\mu$, which we will relate to energy and momentum, and $\varphi$, which we will relate to charge; they are essentially the hydrodynamic degrees of freedom. Supposing we have successfully integrated out all the non-hydrodynamic modes, the generating function may be written as a path integral of the effective action:

$$e^{W[e^\alpha_{s,\mu}, A_{s,\mu}]} = \int DX_1 DX_2 D\varphi_1 D\varphi_2 \ e^{iI_{\mathrm{EFT}}[e^\alpha_{1,M}, B_{1,M}; e^\alpha_{2,M}, B_{2,M}]}. \tag{5.3}$$

To incorporate diffeomorphism invariance and to promote the coordinate fields $X^\mu$ to be dynamical, it is helpful to introduce another fluid spacetime parametrized by $\sigma^M$, as we will illustrate in the following. In the above equations, $M, N, \ldots = t, 1, 2, 3$ represent fluid spacetime indices, while $I, J, \ldots$ represent fluid spatial indices only. In general, the building-blocks $e^\alpha_{1,M}$, $B_{s,M}$ are constructed out of the Stueckelberg fields and the background fields located on the fluid spacetime, such that each block is invariant under the spacetime and gauge symmetries.

The helical fluid we consider preserves a twist symmetry $K_z = P_z + \xi^{-1} L_z$, a U(1) charge $Q$, and the translational symmetries in the $x$-$y$ plane $P_x, P_y$ as well as in time $P_0$. All the other continuous symmetries, including rotations and boosts, are *explicitly* broken. By coupling to background gauge fields, the vielbein[2] $e^\alpha_\mu$ and the U(1) gauge field $A_\mu$, the invariant blocks on the Schwinger-Keldysh contours are given by ($s = 1, 2$)

$$e^0_{s,M}(\sigma) = \partial_M X^\mu_s(\sigma) e^0_{s,\mu}(\sigma), \tag{5.4a}$$

$$e^{b'}_{s,M}(\sigma) = \partial_M X^\mu_s(\sigma) e^{c'}_{s,\mu}(\sigma) R_{s,c'}{}^{b'}(\sigma), \tag{5.4b}$$

$$e^z_{s,M}(\sigma) = \partial_M X^\mu_s(\sigma) e^z_{s,\mu}(\sigma), \tag{5.4c}$$

$$B_{s,M}(\sigma) = \partial_M \varphi_s(\sigma) + \partial_M X^\mu_s A_{s,\mu}(\sigma), \tag{5.4d}$$

where

$$R_{s,c'b'} = \cos\left(\xi^{-1} X^z_s\right) \delta_{c'b'} - \sin\left(\xi^{-1} X^z_s\right) \epsilon_{c'b'z}. \tag{5.5}$$

This rotation matrix is demanded by the twist symmetry $K_z$ in a way that rotations in the internal space are supplemented by a (opposite) translation along the $z$-direction.[3] For the sake of conciseness, we denote

$$\bar{e}^\alpha_{s,\mu}(\sigma) = \delta^\alpha_0 e^0_{s,\mu}(\sigma) + \delta^\alpha_{b'} e^{c'}_{s,\mu}(\sigma) R_{s,c'}{}^{b'}(\sigma) + \delta^\alpha_z e^z_{s,\mu}(\sigma). \tag{5.6}$$

It is convenient to introduce the $r, a$-fields as following

$$\Lambda_r = \frac{\Lambda_1 + \Lambda_2}{2}, \tag{5.7a}$$

---

[2] For our purpose, the vielbein is enough to source energy-momentum tensors in the flat spacetime limit. However, to consider a general curved spacetime, one needs to include spin connections: see e.g. brief remarks in [50].

[3] We have also used the coset construction [51] to build these invariant building blocks. However, this is a bit delicate, because one needs to choose the symmetry generator to be $P_z - \xi^{-1} L_z$. The relative sign difference seems to arise because the rotation of the operators in the coset construction acts as a passive rather than active transformation.

$$\Lambda_a = \Lambda_1 - \Lambda_2, \tag{5.7b}$$

where $\Lambda_{r,a}$ denote collectively the background and dynamical fields. It is clear that (5.4) with suppressed $s$ indices gives the $r$-fields of the blocks. We will see that the $r$-fields will correspond to hydrodynamic degrees of freedom while $a$-fields correspond to fluctuations and noise. Below, we omit the index $r$ for simplicity. In the classical limit and physical spacetime, the $a$-fields of the blocks in (5.4) can be written as

$$e_{a,M}^\alpha = \partial_M X^\mu \bar{E}_{a,\mu}^\alpha, \quad \bar{E}_{a,\mu}^\alpha = \bar{e}_{a,\mu}^\alpha + \mathcal{L}_{X_a} \bar{e}_\mu^\alpha, \tag{5.8a}$$

$$B_{a,M} = \partial_M X^\mu C_{a,\mu}, \quad C_{a,\mu} = A_{a,\mu} + \partial_\mu \varphi_a + \mathcal{L}_{X_a} A_\mu, \tag{5.8b}$$

where, for us, the classical limit schematically corresponds to the $a$-fields being perturbatively small. In the above equations, we have also used the Lie derivative $\mathcal{L}_{X_a}$ along $X_a^\mu$. We turn on a non-zero electromagnetic field $A_\mu$, but take the flat spacetime limit $e_\mu^\alpha = \delta_\mu^\alpha$.[4] We retain the $a$-fields of the background fields. The resulting vielbein $a$-fields become

$$\bar{E}_{a,\mu}^0 = e_{a,\mu}^0 + \partial_\mu X_a^0, \tag{5.9a}$$

$$\bar{E}_{a,\mu}^{b'} = e_{a,\mu}^{c'} R_{c'}^{b'} - \delta_\mu^{c'} \epsilon_{c'd'z} R^{d'b'} \xi^{-1} X_a^z + \partial_\mu X_a^{c'} R_{c'}^{b'}, \tag{5.9b}$$

$$\bar{E}_{a,\mu}^z = e_{a,\mu}^z + \partial_\mu X_a^z. \tag{5.9c}$$

Besides the spacetime and gauge symmetries, we further require the effective action to be invariant under relabeling symmetries, which distinguish the fluid from other phases of matter including solids and superfluids, where a symmetry is spontaneously broken:

$$g(\sigma) \to g(\sigma^M + \xi^M(\sigma^I)), \tag{5.10a}$$

$$g(\sigma) \to g(\sigma) e^{i\lambda_Q(\sigma^I)Q}, \tag{5.10b}$$

where $\xi^M, \lambda_Q$ are arbitrary functions of spatial fluid coordinates only. Since the transformation does not depend on the S-K contour, all the $a$-fields are invariant under relabeling symmetries. However, the invariant $r$-fields without derivatives are only $e_t^\alpha$ and $B_t$. These $r$-fields hence define the thermodynamic variables $\beta$, $u^b$ and $\mu$ as

$$\beta u^\mu = \partial_t X^\mu, \quad u^b = u^\mu e_\mu^b, \quad \beta\mu = B_t. \tag{5.11}$$

There are some remaining constraints on the generating function in terms of the path integral formalism: unitarity and stability of the effective action require

$$I_{\text{EFT}}^*[\Lambda_a, \Lambda_r] + I_{\text{EFT}}[-\Lambda_a, \Lambda_r] = 0, \tag{5.12a}$$

$$\text{Im } I_{\text{EFT}} \geq 0, \tag{5.12b}$$

$$I_{\text{EFT}}[\Lambda_a = 0, \Lambda_r] = 0, \tag{5.12c}$$

Therefore, we can already write down the most general effective action $I_{\text{EFT}} = \int d^4 x \mathcal{L}_{\text{cl}}$ to linear order in $a$-fields and in the classical limit and physical spacetime:

$$\mathcal{L}_{\text{cl}} = \bar{T}_\alpha^\mu \bar{E}_{a,\mu}^\alpha + J^\mu C_{a,\mu} + \ldots = T_\alpha^\mu \left( e_{a,\mu}^\alpha + \partial_\mu X_a^\nu \delta_\nu^\alpha \right) + J^\mu C_{a,\mu} - \Gamma X_a^z + \ldots, \tag{5.13}$$

where $T_\alpha^\mu = \delta_\alpha^\nu \bar{e}_\nu^\beta \bar{T}_\beta^\mu$ is the stress tensor, $J^\mu$ is the current, and

$$\Gamma = T_{b'}^\mu \delta_{\mu c'} \epsilon^{c'b'z} \xi^{-1}. \tag{5.14}$$

Note that the coefficient $\Gamma$ is completely fixed by the anti-symmetric part of the stress tensor, which is nonzero only when $L_z$ is explicitly broken. The variation of (5.13) with respect to the dynamical fields gives the Ward idensities

$$\partial_\mu T_\alpha^\mu \delta_\nu^\alpha = F_{\nu\mu} J^\mu - \Gamma \delta_{\nu z}, \tag{5.15a}$$

$$\partial_\mu J^\mu = 0, \tag{5.15b}$$

where $F = dA$ is the field strength. If we can write $\Gamma = \gamma \pi_z, \gamma > 0$, then the $z$-momentum would be relaxed.

In the following, we will construct the effective action for both the ideal and the first-order hydrodynamics of the helical fluids. In principle, our effective field theory also accounts for the non-linear effects but the full analysis will not be discussed here.

---

[4] Although we work in the flat limit, we keep all the vielbein indices exact, so a generalization to curved spacetime is straightforward.

## 5.2. Ideal hydrodynamics

Ideal hydrodynamics in equilibrium is described by variational principles of an effective action as a function of invariant blocks [55]. The most general equilibrium action with single time is given by

$$S_0 = \int \mathrm{d}^4 x e P(e_t^\alpha, B_t), \tag{5.16}$$

where $e = \det e_\mu^\alpha$. Then, the variation with respect to the background gauge fields produces the stress tensor and currents,

$$T_{(0)0}^\mu = -\varepsilon u^\mu - p(u^\mu - e_0^\mu), \tag{5.17a}$$

$$T_{(0)b}^\mu = p e_b^\mu + \pi_b u^\mu, \tag{5.17b}$$

$$J_{(0)}^\mu = n u^\mu, \tag{5.17c}$$

where the equation of state is given by

$$p = P, \quad \varepsilon = -\beta \frac{\partial P}{\partial e_t^0} - p, \quad \pi_b = \beta \bar{e}_\mu^c e_b^\mu \frac{\partial P}{\partial e_t^c}, \quad n = \beta \frac{\partial P}{\partial B_t}, \tag{5.18}$$

and all the partial derivatives are taken with other arguments being fixed. To proceed, we vary the action with respect to the dynamical fields (in the flat spacetime limit):

$$\delta S_0 \simeq P \partial_\mu \delta X^\mu + \beta \frac{\partial P}{\partial e_t^0} u^\mu \partial_\mu X^0 + \beta \frac{\partial P}{\partial e_t^c} \bar{e}_i^c u^\mu \partial_\mu X^i - \beta u^{b'} \epsilon_{b'}^{i'z} \bar{e}_{i'}^{c'} \frac{\partial P}{\partial e_t^{c'}} \xi^{-1} \delta X^z$$
$$+ \beta \frac{\partial P}{\partial B_t} u^\mu \partial_\mu \delta \varphi + \beta \frac{\partial P}{\partial B_t} u^\mu F_{\nu\mu} \delta X^\nu. \tag{5.19}$$

The resulting Ward identity is identical to (5.15) with

$$\Gamma_{(0)} = \pi_{b'} u^{i'} \epsilon_{i'}^{b'z} \xi^{-1}. \tag{5.20}$$

At equilibrium, we can write $\pi_{b'} = \rho_\perp u_{b'}$ with $\rho_\perp$ the momentum susceptibility in the $x$-$y$ plane, hence $\Gamma_{(0)} = 0$. However, as we saw in kinetic theory in the previous sections, the rotational viscosity would give rise to an anti-symmetric stress tensor that, according to (5.14), renders $\Gamma_{(0)} \neq 0$. Unlike conventional fluids (with discrete rotational symmetry), the part of the rotational viscosity for the helical fluid carries factor $\xi^{-1}$ instead of spatial derivatives $\sim \ell^{-1}$. Recall that we are always working in the regime where $\ell \gg \xi$, therefore, the finite $\Gamma_{(0)}$ indeed guarantees that there is no propagating modes along $z$-direction. Since the rotational viscosity is a dissipative effect, we find it is convenient to present the results in Section 5.3, even though (as we saw above) it will ultimately relax $v_z$ (and thus give rise to terms at "$-1$ derivative order" in the gradient expansion).

Finally, we point out the relation between the single-time action and the S-K formalism at equilibrium. The effective actions are related by

$$I_{\text{EFT,eq}} = S_0[\Lambda_1] - S_0[\Lambda_2], \tag{5.21}$$

which is known as factorizability [48–50]. In the classical limit and physical spacetime, one would arrive at (5.13) with the coefficients given in (5.17) and (5.20). However, to go from the S-K formalism to the factorized action, one needs to be careful to avoid certain forbidden terms by coupling to a generic background field [50].

## 5.3. First order hydrodynamics

In general (at higher orders in the derivative expansion), the effective action is not factorizable,

$$I_{\text{EFT}} = S_0[\Lambda_1] - S_0[\Lambda_2] + \text{(higher derivative terms)}, \tag{5.22}$$

To write down the higher derivative terms, we notice another symmetry on the S-K contour: when $\rho_0 = \mathrm{e}^{-\beta_0 H}$ is taken to be the thermal ensemble, the Kubo-Martin-Schwinger (KMS) condition [48, 49] tells that $I_{\text{EFT}}$ is related to its time-reversal partner with every field getting an imaginary shift along the temporal direction. By further

applying an anti-unitary symmetry $\Theta$ that is preserved by the Hamiltonian, we obtain back the original action $I_{\text{EFT}}[\Lambda_1, \Lambda_2] = I_{\text{EFT}}[\tilde{\Lambda}_1, \tilde{\Lambda}_2]$, where tilde represents the KMS transformation. For the helical fluid, we take $\Theta = \mathcal{I}_\perp \mathcal{T}$, where $\mathcal{T}$ is the time-reversal symmetry and $\mathcal{I}_\perp$ is the inversion symmetry within the $x$-$y$ plane.[5] Then, the KMS transformation is given by

$$\tilde{\bar{e}}_\mu^\alpha(\Theta x) = \Theta \bar{e}_\mu^\alpha(x), \tag{5.23a}$$

$$\tilde{\bar{E}}_{a,\mu}^\alpha(\Theta x) = \Theta \bar{E}_{a,\mu}^\alpha(x) + i\Theta \mathcal{L}_\beta \bar{e}_\mu^\alpha, \tag{5.23b}$$

$$\tilde{B}_\mu(\Theta x) = \Theta B_\mu(x), \tag{5.23c}$$

$$\tilde{C}_{a,\mu}(\Theta x) = \Theta C_{a,\mu}(x) + i\Theta \mathcal{L}_\beta B_\mu, \tag{5.23d}$$

where $\beta^\mu = \beta u^\mu$, and the $r$-fields are taken from (5.4) with $\sigma^M = X^\mu \delta_\mu^M$. Under $\Theta$, various fields transform as the following,

$$x^\mu \to (-x^0, -x^{i'}, x^z), \tag{5.24a}$$

$$u^\mu \to (u^0, u^{i'}, -u^z), \tag{5.24b}$$

$$\partial_\mu \to (-\partial_0, -\partial_{i'}, \partial_z), \tag{5.24c}$$

$$A_\mu \to (A_0, A_{i'}, -A_z), \tag{5.24d}$$

$$e_\mu^\alpha \to (e_0^0, e_0^{b'}, e_{i'}^0, e_{i'}^{b'}, -e_0^z, -e_{i'}^z, -e_z^0, -e_z^{b'}, e_z^z). \tag{5.24e}$$

The Lie derivatives in (5.23) can be written explicitly as

$$\mathcal{L}_\beta \bar{e}_\mu^\alpha = \partial_\mu \beta^\rho \bar{e}_\rho^\alpha - \delta_{b'}^\alpha \delta_\mu^{i'} \bar{e}_{i'c'} \epsilon^{c'b'z} \xi^{-1} \beta^z, \tag{5.25a}$$

$$\mathcal{L}_\beta B_\mu = \partial_\mu(\beta\mu) + \beta^\nu F_{\nu\mu}. \tag{5.25b}$$

Here we keep $\xi^{-1}$ in the first derivative order to ensure that all the non-hydrodynamic modes would be integrated out properly. Again, in the following, we will work in the classical limit and physical flat spacetime.

The most general $\mathcal{I}_\perp \mathcal{T}$-even Lagrangian containing two factors of $a$-fields and zero derivatives is

$$-i\beta \mathcal{L}_{\text{cl,even}}^{(2,0)} = \sigma^{ij} C_{a,i} C_{a,j} + \kappa^{ij} \bar{E}_{a,i}^0 \bar{E}_{a,j}^0 + 2\nu^{ij} C_{a,i} \bar{E}_{a,j}^0 + s^{ijkl} \bar{E}_{a,ij} \bar{E}_{a,kl}, \tag{5.26}$$

where $\bar{E}_{a,ij} = \bar{E}_{a,i}^b e_{bj}$ and the invariant tensors are

$$\sigma^{ij} = \sigma_\perp \delta^{i'j'} + \sigma_\parallel \delta_z^i \delta_z^j, \quad \kappa^{ij} = \kappa_\perp \delta^{i'j'} + \kappa_\parallel \delta_z^i \delta_z^j, \quad \nu^{ij} = \nu_\perp \delta^{i'j'} + \nu_\parallel \delta_z^i \delta_z^j,$$

$$s^{ijkl} = \zeta_\perp \delta^{i'j'} \delta^{k'l'} + \zeta'(\delta^{i'j'} \delta_z^k \delta_z^l + \delta_z^i \delta_z^j \delta^{k'l'}) + \zeta_\parallel \delta_z^i \delta_z^j \delta_z^k \delta_z^l$$

$$+ \eta_\perp \delta^{i'<k'} \delta^{l'>j'} + \eta_1' \delta_z^i \delta_z^k \delta^{l'j'} + \eta_2' \delta^{i'k'} \delta_z^l \delta_z^j + \eta_3'(\delta_z^i \delta_z^l \delta^{j'k'} + \delta_z^j \delta_z^k \delta^{i'l'}) + \eta_\circ \epsilon^{i'j'z} \epsilon^{k'l'z}, \tag{5.28}$$

with $A^{<ij>} = A^{ij} + A^{ji} - \frac{2}{d} \delta^{ij} A^{kk}$. From the unitarity and stability constraints (5.12), we have the *dissipative* coefficients

$$\sigma_{\perp,\parallel}, \kappa_{\perp,\parallel}, \zeta_{\perp,\parallel}, \zeta', \eta_{\perp,\parallel}, \eta_{1,2,3}', \eta_\circ \geq 0, \quad \nu_{\perp,\parallel}^2 \leq \sigma_{\perp,\parallel} \kappa_{\perp,\parallel}. \tag{5.29}$$

It is clear that the dissipative effects are similar to the normal fluids with $\sigma, \kappa, \nu$ being the thermoelectric coefficients and $\zeta, \eta$ being the bulk and shear viscosities; while, we allow extra transport coefficients, e.g. the rotational viscosity $\eta_\circ$. As usual, the terms proportional to $E_{a,t}^\alpha$ and $C_{a,t}$ can be eliminated by field redefinition [49]. Then, the KMS condition requires that at linear order in $a$-fields, $2\mathcal{L}_{\text{cl,odd}}^{(1,1)} = \widetilde{\mathcal{L}}_{\text{cl,even}}^{(2,0)}|_{\mathcal{O}(a)}$. This leads to

$$\beta \mathcal{L}_{\text{cl,odd}}^{(1,1)} = -\sigma^{ij} \mathcal{L}_\beta B_i C_{a,j} - \kappa^{ij} \mathcal{L}_\beta \bar{e}_i^0 \bar{E}_{a,j}^0 - \nu^{ij} \mathcal{L}_\beta B_i \bar{E}_{a,j}^0 - \nu^{ij} \mathcal{L}_\beta \bar{e}_i^0 C_{a,j} - s^{ijkl} \mathcal{L}_\beta \bar{e}_i^b e_{bj} \bar{E}_{a,kl}. \tag{5.30}$$

Moreover, there exists a $\mathcal{I}_\perp \mathcal{T}$-even Lagrangian at $\mathcal{O}(a)$ that remains invariant under the KMS transformation. This is given by

$$\beta \mathcal{L}_{\text{cl,even}}^{(1,1)} = \lambda_1^{i,jk} \left( \mathcal{L}_\beta B_i \bar{E}_{a,jk} - C_{a,i} \mathcal{L}_\beta \bar{e}_j^b e_{bk} \right) + \lambda_2^{i,jk} \left( \mathcal{L}_\beta \bar{e}_i^0 \bar{E}_{a,jk} - \bar{E}_{a,i}^0 \mathcal{L}_\beta \bar{e}_j^b e_{bk} \right), \tag{5.31}$$

---

[5] Note that $\mathcal{I}_\perp$ and $\mathcal{T}$ are independently symmetries of the system. The choice here is convenient as it means that most of the variables in (5.23) will not pick up a minus sign under KMS.

where

$$\lambda_1^{i,jk} = \alpha_1 \delta_z^i \epsilon^{zjk} + \beta_1 \delta_z^j \epsilon^{izk} + \bar{\beta}_1 \delta_z^k \epsilon^{ijz}, \quad \lambda_2^{i,jk} = \alpha_2 \delta_z^i \epsilon^{zjk} + \beta_2 \delta_z^j \epsilon^{izk} + \bar{\beta}_2 \delta_z^k \epsilon^{ijz}. \tag{5.32}$$

Note that the $z$-direction, which is invariant under $\mathcal{I}_\perp \mathcal{T}$, provides the opposite parity of the Lagrangian. The coefficients $\alpha_{1,2}$, $\beta_{1,2}$ and $\bar{\beta}_{1,2}$ are unconstrained and represent *dissipationless* coefficients; they separately indicate whether the vector ($\alpha$) or the tensor ($\beta$) has an index in the $z$-direction.

Comparing (5.30) and (5.31) to (5.13), we arrive at the first-order stress tensor and currents:

$$T_{(1)0}^i = -\kappa^{ij}\beta^{-1}\partial_j\beta - \nu^{ij}\beta^{-1}\partial_j(\beta\mu) - \alpha_2\delta_z^i\left(\omega - 2\xi^{-1}u^z\right) - \beta_2\epsilon^{ijz}\beta^{-1}\partial_j\beta^z - \bar{\beta}_2\epsilon^{izk}\beta^{-1}\partial_z\beta_k, \tag{5.33a}$$

$$\begin{aligned} T_{(1)b}^i = &-s^{kli}{}_b\beta^{-1}\partial_k\beta_l + s^{k'l'i}{}_b\epsilon_{k'l'z}\xi^{-1}u^z \\ &+ \beta^{-1}\left[\left(\alpha_1\partial_z(\beta\mu) + \alpha_2\partial_z\beta\right)\epsilon^i{}_{bz} + \left(\beta_1\partial_k(\beta\mu) + \beta_2\partial_k\beta\right)\delta_z^i\epsilon^{kz}{}_b + \left(\bar{\beta}_1\partial_k(\beta\mu) + \bar{\beta}_2\partial_k\beta\right)\delta_{bz}\epsilon^{kiz}\right], \end{aligned} \tag{5.33b}$$

$$J_{(1)}^i = -\nu^{ij}\beta^{-1}\partial_j\beta - \sigma^{ij}\beta^{-1}\partial_j(\beta\mu) - \alpha_1\delta_z^i\left(\omega - 2\xi^{-1}u^z\right) - \beta_1\epsilon^{ijz}\beta^{-1}\partial_j\beta^z - \bar{\beta}_1\epsilon^{izk}\beta^{-1}\partial_z\beta_k, \tag{5.33c}$$

where to simplify the notations, we have taken $x^z = 0$, such that $\bar{e}_\mu^\alpha = e_\mu^\alpha$, and we defined the vorticity as

$$\omega = \epsilon^{zj'k'}\beta^{-1}\partial_{j'}\beta_{k'}. \tag{5.34}$$

Moreover, we obtain the first-order $\Gamma$

$$\Gamma_{(1)} = 2\left[-\eta_\circ(\omega - 2\xi^{-1}u^z) + \alpha_1\beta^{-1}\partial_z(\beta\mu) + \alpha_2\beta^{-1}\partial_z\beta\right]\xi^{-1}. \tag{5.35}$$

Importantly, the term proportional to $\eta_\circ$ will relax $\pi_z$. From $\Gamma = \gamma\pi_z$, we can read out the relaxation time $\gamma^{-1}$, given by (3.29), where $\rho_\parallel$ is the momentum susceptibility along $z$-direction. Therefore, if the time scale we are interested in satisfies $t \gg \gamma^{-1}$, there will be no propagating mode along $z$-direction. This then justifies the claim below (5.20). Hence, together with $\ell \gg \xi$, we expect that these two criteria define the regime of validity of our helical fluids: see the corresponding discussion at the end of Section 4.

As in Section 4, the equation of motion for $u^z$ becomes approximately

$$\partial_i T_z^i = -\Gamma_{(1)} \tag{5.36}$$

on long time scales $t \gg \gamma^{-1}$. We conclude that

$$u^z = \frac{\xi}{2}\omega + c_1\partial_z\mu + c_2\partial_z\beta + \cdots, \tag{5.37}$$

where $\cdots$ denotes higher derivative terms, and the constants $c_{1,2}$ are unimportant for what follows. We can now plug in this constraint equation for $u_z$ into the constitutive relations, and we find that

$$J^z = n\frac{\xi}{2}\omega + \cdots, \tag{5.38a}$$

$$T_0^z = -(\epsilon + p)\frac{\xi}{2}\omega + \cdots. \tag{5.38b}$$

The coefficient in $J^z$ is exactly the same as what we found before in (3.14), while the energy current $T_0^z$ has an analogous coefficient.

It is especially interesting to study the antisymmetric contribution to the stress tensor. This is nicely read off by

$$T_y^x - T_x^y = \xi\Gamma_{(1)} = -\xi\partial_i T_z^i = -\xi\partial_z p + \cdots. \tag{5.39}$$

Again the $\cdots$ denotes higher derivative terms. This result provides the advertised generalization of the discussion in Section 3.4: a pressure gradient along the helix of a helical fluid will generically cause a torque. We hope that this effect can be detected in experiment.

## 6. CONCLUSION

In this work, we have described the exotic hydrodynamics of materials with explicit helical symmetry. In addition to offering a fruitful playground to explore complementary perspectives on hydrodynamics, the fluid with helical symmetry group displays several new features and transport coefficients that make searches for experimental realizations

of it worthwhile. First, despite still displaying a three-dimensional flow pattern, the helical fluid's velocity field is two dimensional. It would be interesting to understand the fate of turbulent flows in such fluids, and it seems plausible that the character of turbulence will be closer to two-dimensional than three-dimensional [56]. Secondly, the helical fluid features transverse responses to external fields, in particular supplying torque in the presence of a pressure gradient: see (5.39). These effects are in principle quite simple and experimentally testable in principle, although early authors [6] neglected them. The magnitude of the effect may be quite small.

One promising candidate for realizing explicit helical order is a thin film of *cholesteric liquid crystals* [31, 32]. For these materials, which *spontaneously* choose a helix axis, appropriate pinning of the thin film may align the helical axis in a chosen direction. And while there is an additional hydrodynamic mode corresponding to the propagating Goldstone mode associated with the sliding of the helix, we expect that a transverse magnetic field would gap out this Goldstone mode. (Note that in these liquid crystal films it is *mass*, not electric charge, which will play the role of a scalar conserved quantity; therefore this magnetic field will not lead to a strong cyclotron response.)

While the symmetry pattern discussed in this work is *continuous* helical symmetry, there are also interesting systems where this continuous helical translation symmetry is broken to a discrete subgroup. One example is a crystal such as KHgSb [57] with a non-symmorphic point group, where a translation along one crystal axis must be combined with a discrete rotation in the plane to be a symmetry. The novel effects described in this paper must also be present for such "discrete helical symmetry", although there could be even further effects we did not find in this paper arising from the lower rotational symmetry.

To close, the helical symmetry pattern displays rich behavior beyond the well-known hydrodynamics of Galilean fluids. We look forward to future experimental and theoretical investigations of the consequences of space-time symmetry mixing in hydrodynamics.

### ACKNOWLEDGMENTS

We thank Leo Radzihovsky for helpful discussions. This work was supported in part by the Alfred P. Sloan Foundation through Grant FG-2020-13795 (AL), the National Science Foundation through CAREER Grant DMR-2145544 (XH, AL), and through the Gordon and Betty Moore Foundation's EPiQS Initiative via Grant GBMF10279 (JF, XH, AL).

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
