# Peer review of "Hydrodynamics with helical symmetry"

_SciPost Physics_

## Round 1 · Referee Report · Anonymous (Referee 3) · 2022-10-21

Report

The paper aims to develop the hydrodynamic theory for a fluid which has translations along and rotations about the $z$-axis explicitly broken down to a helical symmetry. The authors present their hydrodynamic formulation using the standard Landau paradigm, where they incorporate all possible terms in the gradient expansion which are consistent with the symmetries of their setup. They also provide arguments based on kinetic theory in a simple model for why the effects of rotational viscosity can be ignored, and how the effective motion of the fluid becomes two-dimensional. Finally, they also appeal to the Schwinger-Keldysh closed time path formulation for non-equilibrium systems to provide further justification for their helical fluid construction.

Unfortunately, several aspects of their physical setup and approach are not very clear, due to which I can not recommend the publication of the manuscript in its current form. The authors should carefully address the following concerns about the manuscript before it can be considered for publication.

1. Rotations in the $x-y$ plane are one of the symmetries of the system, as discussed in section 2, where they are denoted via the matrix $R_{AB}$, and form the $SO(2)$ part of the full symmetry group. However, $L_z$ is not conserved, except through the combination $K_z \equiv P_z + \xi^{-1} L_z$. The two statements appear mutually contradictory. If rotations in the $x-y$ plane are a continuous symmetry, then the associated generator of such rotations, $L_z$, should be conserved. The authors should comment upon and clear up this apparent ambiguity. This can potentially conflict with the authors' counting of the independent invariant tensors in terms of which the hydrodynamic gradient expansion has been constructed.

2. In section 3.1, the authors assume $\boldsymbol{\pi} = n_0 \boldsymbol{v}$. This needs some justification. The proportionality between momentum density and mass flux in a non-relativistic theory is a consequence of two things: (i) Translation invariance, and (ii) Galilean boost invariance (which leads to the conservation of the centre of mass). The authors do assume translation invariance along $x,y$ directions, but do not assume boost invariance. If they also assume Galilean boost invariance along $x,y$ directions, then $\pi_x = n_0 v_x$ and $\pi_y = n_0 v_y$ will hold true.
The situation with the z-axis is even more complicated. Here the authors have neither translation invariance nor boost invariance. Thus $\pi_z \neq n_0 v_z$, which is in conflict with the subsequent analysis the authors do. The authors should clarify these issues.

3. What is the motivation behind writing eq. (3.6)? In other words, why should one not have the more general structure $D_{iAkB} = \eta_1 \delta_{ik} \delta_{AB} + \eta_2 \delta_{iB} \delta_{Ak} +\eta_3 \delta_{iA} \delta_{kB}$, with three independent transport parameters? There need not be an $i \leftrightarrow A$ exchange symmetry, since non-conservation of $L_z$ implies $\tau_{xy} \neq \tau_{yx}$.

4. The notation in eq. (3.8) is opaque. What do the $\epsilon$ symbols with only two indices mean? Also, why can one not write this term more generally as $D_{iAkB} = \eta_o^{(1)} \epsilon_{iAj} \epsilon_{jkB} + \eta_o^{(2)} \epsilon_{ikj} \epsilon_{jAB}$ i.e. with two different rotational viscosities?

5. The premise surrounding the calculations in eqs. (3.9) - (3.11) is unclear. Why should one assume $v_z = 0$? Even if one does, since $\pi_z \neq n_0 v_z$ (see point 2 above), why should vanishing of $v_z$ imply vanishing of $\pi_z$, and hence conservation of $L_z$?

6. In section 3.2, the authors try to provide a justification for setting $v_z = 0$. However this also inherently assumes $\pi_z \propto v_z$. The key equation which the authors use to justify relaxation of $v_z$ is eq. (3.28). This is, however, far from clear. The RHS of eq. (3.28) involves not just ${v_z}/{\xi^2}$, but also the term $\omega_z/\xi$. In fact, in the limit of large $\xi$, where one would expect the translation invariance along the z-axis to be almost restored, it is the term $\omega_z/\xi$ that will dominate over $v_z/\xi^2$. This couples the time evolution of $\pi_z$ with that of $v_x, v_y$, which could be quite non-trivial, and will not in general lead to a relaxation of $v_z$. The authors perhaps inherently assume a scaling for spatial derivatives of the form $\partial \sim \xi^{-1}$. If so, then such an assumption should be clearly mentioned.

7. The parameters $\alpha, a$, present in eqns. (3.13a), (3.14a), are proportional to $\xi$, as concluded in eq. (3.31). This is once again counter-intuitive. In the limit of large $\xi$, where translations and rotations get restored, one would expect these terms to almost vanish. In fact, by setting $\xi \rightarrow \infty$, one should expect $\tau_{xy} - \tau_{yx}$ to vanish, purely due to spatial isotropy. But the conclusion that $\alpha \propto \xi$ seems to be entirely in violation of this physical expectation.

Similar issues arise with the computation of torque on an object in section 3.4, which blows up in the limit of large $\xi$, which one would instead have expected to diminish.

8. Once the authors have assumed relaxation of $v_z$, why is it then required to include eq. (3.33) into the analysis of small fluctuations about the equilibrium state?

9. The authors introduce several novel transport parameters in section 3. However, they do not perform a systematic entropy current analysis, which can shed further light on these parameters. By performing an entropy current analysis, the authors should ensure that none of the new parameters they introduce leads to a violation of the second law of thermodynamics i.e. the positivity of entropy production.

10. In section 4.3, the authors argue that the limit $\xi \rightarrow \infty$ gives unphysical results because helical hydrodynamics breaks down. But from their symmetry based arguments in section 3, the limit $\xi \rightarrow \infty$ is the one in which ordinary hydrodynamics should emerge. Thus one would once again expect that in this limit, any transport parameters that exist purely in helical hydrodynamics should tend to vanish. However, this is not the behaviour observed for transport parameters such as $\alpha$ (point 7 above). The authors should carefully chalk out the emergence of ordinary hydrodynamics from helical hydrodynamics after symmetry restoration in the limit $\xi\rightarrow \infty$.

11. In section 5, why is it justified to have a flat spacetime background rather than one which naturally only has the helical symmetry intact? If the authors want to consider an explicit breaking of $P_z, L_z$ down to the helical symmetry $K_z$, should not there be something external to the fluid, such as the background metric, which enforces this explicit symmetry breaking pattern?

12. In eq. (5.13), the authors add the term $\Gamma X^z_a$ by hand to make a momentum-relaxing term appear in the hydrodynamic equation (5.15a). How will the field redefinitions of $X^z_a$ affect their conclusions?

---

## Round 1 · Referee Report · Anonymous (Referee 1) · 2023-4-17

Report

In addition to the other reports, I think it is important that the authors connect to the relevant literature from the AdS/CFT community. For instance, in http://arxiv.org/abs/1406.6351 and http://arxiv.org/abs/1412.3446, the dc electric and thermal conductivities were computed in phases with a holographic dual endowed with a helical symmetry. From these results, it should be possible to extract the expression for the momentum relaxation rate in the direction of the helix and compare it to the formula the authors of the present manuscript have derived using hydrodynamic considerations. Do the two results agree?

---

## Round 1 · Referee Report · Anonymous (Referee 2) · 2023-4-17

Report

The paper addresses the problem of constructing a hydrodynamic theory for systems that microscopically are invariant under translations in one plane and a twisted translation along the transverse direction dubbed helical symmetry. In the first part of the paper, they introduce the symmetry and construct a set of hydrodynamic constitutive relations using the symmetry-based hydrodynamic paradigm, although second law of thermodynamics is not imposed. Then, in the second half, they use kinetic theory and construct a generating function using Schwinger-Keldysh formalism as crosscheck and gain more insight on this exotic problem.

After reading the paper and the first referee report, I agree several aspects must be clarified.

In section 2, the authors introduce the helical symmetry group. The whole discussion should probably be reorganised. They focus on the properties of the group $O(2)$ and how acts on the generators, which seems confusing because $O(2)$ is not the group they are considering. Notice $O(2)$ is a compact group whereas the helical group is not. In addition, the origin of $\sigma$ transformations as part of the helical group is not properly motivated. Is the $\sigma$ transformation introduced because the helix remains invariant after a $\pi$ rotation around the $y$ axis followed by a $\pi$ rotation around the $z$ axis?

In the hydrodynamic section, the authors argue that the hydro description will be captured just by the conservation equations of charge and transverse momentum. However, there is an extra conserved quantity in the problem, why is that equation not considered a hydrodynamic equation? Somehow the nonrigorous discussion around 3.9 and subsection 3.2 is secretly using that extra conservation law.

As the first referee argues the definition $\boldsymbol \pi=n\bf v$ introduced below equations 3.3 is not justified due to the lack of Galilean invariance. On the other hand, below equation 3.13 it is used $\boldsymbol \pi=mn\bf v$, and in section 5 it is used $\boldsymbol \pi=\rho\bf v$. It would be convenient to have unified notation.

Why is $\pi_z$ included in the Onsager analysis?

I do not understand the argument after 3.20 $v_z$ and $\Omega_z$ are not thermodynamic meaningful quantities, in fact, they do not appear in the density matrix 3.21. Given the symmetry algebra and that density matrix, I do not see how equations 3.22-3.23 can be obtained. How are frames defined and related between them? Under translations with parameter $\lambda^A$, the chemical potentials will transform as $v'_A\sim v_A+w\epsilon_{AB}\lambda^B/\xi$, and under twisted translations $v'_A\sim R_{AB}(\lambda_z/\xi)v^B$. Therefore, all the analyses and conclusions in this section seem not well justified.

In equation 3.25 it is claimed $\tau_{xy}-\tau_{yx}$ is "boost" invariant without proof. What do they mean by boost invariance?

Why is $\pi_z$ just proportional to $v_z$? For example, in their treatment, $\omega_z$ is of the same order in the derivative expansion as $v_z$ and possesses the same transformation property under $\sigma$.

Can the authors clarify the relation between the chiral vortical effect and this problem? the terms with $\alpha$ and $\beta$ are there because the symmetry allows them. Actually, in the kinetic theory description, it is argued that these terms will be generically present independently of the microscopic system being made of massless fermions.

Given the criticism of both referees, I do not recommend the paper for publication.

---

## Editorial Decision

awaiting_resubmission